# Advancing the science of dynamic airborne nanosized particles using Nano-DIHM

Devendra Pal [1], Yevgen Nazarenko [1], Thomas C. Preston [1,2] & Parisa A. Ariya [1,2]✉

In situ and real-time characterization of aerosols is vital to several fundamental and applied research domains including atmospheric chemistry, air quality monitoring, or climate change studies. To date, digital holographic microscopy is commonly used to characterize dynamic nanosized particles, but optical traps are required. In this study, a novel integrated digital in-line holographic microscope coupled with a flow tube (Nano-DIHM) is demonstrated to characterize particle phase, shape, morphology, 4D dynamic trajectories, and 3D dimensions of airborne particles ranging from the nanoscale to the microscale. We demonstrate the application of Nano-DIHM for nanosized particles (≤200 nm) in dynamic systems without optical traps. The Nano-DIHM allows observation of moving particles in 3D space and simultaneous measurement of each particle's three dimensions. As a proof of concept, we report the real-time observation of 100 nm and 200 nm particles, i.e. polystyrene latex spheres and the mixture of metal oxide nanoparticles, in air and aqueous/solid/heterogeneous phases in stationary and dynamic modes. Our observations are validated by high-resolution scanning/transmission electron microscopy and aerosol sizers. The complete automation of software (Octopus/Stingray) with Nano-DIHM permits the reconstruction of thousands of holograms within an hour with 62.5 millisecond time resolution for each hologram, allowing to explore the complex physical and chemical processes of aerosols.

[1] Department of Atmospheric and Oceanic Sciences, McGill University, 805 Sherbrooke Street West, Montreal, QC H3A 0B9, Canada. [2] Department of Chemistry, McGill University, 801 Sherbrooke Street West, Montréal, QC H3A 2K6, Canada. ✉email: parisa.ariya@mcgill.ca

In situ and real-time characterization of airborne particle (aerosol) size, phase, and morphology is vital to several fundamental and applied research domains, including atmospheric chemistry and physics[1], air quality[2,3], climate change[4–6], and human health[7]. Aerosols are diverse and are emitted from anthropogenic or natural sources. Aerosols undergo physico-chemical transformations in the atmosphere on a wide range of spatial and temporal scales[3,8]. Aerosols can contain biological particles such as bacteria and viruses[9]. Airborne severe acute respiratory syndrome coronavirus 2 virion-containing aerosols are significant in the coronavirus disease 2019 transmission[10–12].

Nano-objects that comprise nanomaterials refer to matter dispersed into individual objects with one or more external dimensions, or an internal structure, on a scale from 1 to 100 nm[13]. The most abundant airborne particles in the atmosphere are nanosized (<200 nm diameters)[14–16]. Nanosized particles have a large surface-to-volume ratio. They can be photoreactive and serve as cloud condensation nuclei[6,17,18] or ice nuclei[19,20]. Nanoparticles are also involved in coagulation[17,21] and phase transition processes[22], which are vital to understanding aerosol-cloud interactions[5,6,18]. Prior to this study, to our knowledge, the phase of nanoparticles has never been determined in situ in dynamic ambient air.

There has been significant progress in in situ aerosol analysis during the last decade, including in situ analysis of nanoparticles in air[1,2,14–16,23–29]. Yet, no existing technique allows the determination of airborne particles' phase in situ and in real time, and so in situ phase determination, despite its importance, remains a key challenge[30]. There are various real-time aerosol characterization techniques, including in situ optical/laser or condensation-based and electron mobility analysis[16,20,23]. However, these techniques fail to provide information on the behaviour or dimensions of aerosols. Offline analysis following the collection of aerosols using impactors[15,16], impingers, precipitators, and filters is often more informative, yet does not provide real-time data or information regarding aerosol dynamics.

Significant advances in microscopy during the recent decades have enabled researchers to observe individual nanoparticles[31,32], using techniques such as near-field optical microscopy[33–35], super-resolution microscopy[36–38], atomic force microscopy[39], electron microscopy[40], and other more recently developed imaging techniques[41–45]. Among them, scanning transmission electron microscopy (S/TEM) enables acquiring accurate information on particle phase and morphology in stationary mode[20]. Bright-field and dark-field optical microscopy provide two-dimensional (2D) information on particles, albeit in a limited depth interval within samples[42,46]. A challenge with conventional light microscopy methods is that these methods work in fixed imaging planes[46], which precludes determining aerosol dynamics, phase, and three-dimensional (3D) morphology of aerosols[46]. However, the 3D structure information can be obtained using Fourier ptychography[47], optical diffraction tomography[48], or by scanning the whole sample/particle volume using confocal imaging[49]. All these existing microscopy techniques have significant advantages, yet they cannot track moving particles in situ or in real time, precluding their application to dynamic media, such as air.

Here we provide an alternative approach of Nano-DIHM. The Nano-DIHM consists of a holographic microscope and a gas flow tube that allows airborne particles to pass through the imaging volume of the digital in-line holographic microscope (DIHM) and exit or circulate inside the volume, allowing observation of single particles or ensembles of particles in real time. Nano-DIHM directly acquires data on interference patterns of the incident and scattered light with a light-sensitive matrix, without any lenses or objective[50]. The recorded interference pattern referred to as a hologram is numerically reconstructed using an Octopus/Stingray software based on a patented algorithm[51] to recover the object information[50,52,53]. To date, digital holography setups could merely characterize particles held in electrodynamic[54,55] or optical traps[56,57], or particles deposited on a substrate[43,50,53,58–60]. It is noteworthy that optical trapping of airborne particles requires optical tweezers, trapping only a single particle confined to the field of view[57,61,62]. The Nano-DIHM has a larger field of view up to several square centimetres (~1.27 cm², 2048 × 2048 pixels, 5.5 μm each pixel size). In this study, the Nano-DIHM field of view of up to ~40 mm² enables observation of moving objects, in contrast to conventional optical microscopy that uses lenses and has smaller fields of view (a few micrometres)[42,43]. Until now, investigations of airborne particles by DIHM have been confined to relatively large particles (>1 μm)[63–68]. We show that Nano-DIHM can detect nanosized objects in 2D and 3D space for dynamic (air), aqueous (water), and solid (powder) media. We were able to track individual airborne nanoparticles directly and quantify each particle's dimensions in situ and in real time. The Nano-DIHM enabled us to record six-dimensional spatial motion of aerosol particles (3D for the position of each particle in 3D space) and the dimensions of each particle as it is orientated in 3D space[58,69]. A critical advantage of Nano-DIHM is that, during reconstruction, a single 2D hologram can produce a 3D image of the objects without any loss of resolution[50,58]. The Octopus/Stingray software allows real-time or offline reconstruction with a temporal resolution on the order of milliseconds (62.5 ms) and it can be improved to microsecond-scale temporal resolution, depending on the frame rate of the camera[57,65,70,71].

## Results

We demonstrated that the developed Nano-DIHM detects single and cluster aerosols in situ and in real time, in dynamic and stationary media such as air, water, and heterogeneous matrices. We further performed intercomparison of the aerosol size distribution measured using a Scanning Mobility Particle Sizer (SMPS) and an Optical Particle Sizer (OPS) and S/TEM with the observation made by the Nano-DIHM. The SMPS and OPS can only measure particle size distributions and cannot track particle position in space or individual particle dimensions in 3D, in contrast to Nano-DIHM. In the following section, we demonstrate: (1) the observation of airborne polystyrene latex (PSL) particles (100 nm and 200 nm) and their dynamic trajectories; (2) the detection of the 100 nm and 200 nm PSL particles in colloid solution deposited on a microscope slide; (3) distinguishing the PSL and mixed metal oxide nanoparticles; (4) semi-automation of the software (Octopus/Stingray), which allows the reconstruction of thousands of holograms within an hour; and (5) the observation of ambient particles in the air and snow in real time and in situ (Supplementary Note 1). Finally, the demonstration of the accuracy of measurement of refractive index is presented (Supplementary Note 2). The experimental parameters for each experiment are given in Table 1. More details are provided in the 'Methods' section.

## Detection of airborne nanosized objects using Nano-DIHM: beyond the diffraction barrier

One of the significant limitations of optical microscopy, including digital holography, is overcoming the diffraction barrier of the system to detect nanosized objects with visible wavelengths. To mitigate this challenge, recently, different experimental and numerical focusing approaches have been developed[43,60,72–74]. In this study, an ensemble of three routines is used to overcome the diffraction barrier: (1) keeping a minimum distance between the pinhole and the sample, (2) numerical reconstruction using Octopus/Stingray software[52,75] based on Kirchhoff–Helmholtz

**Table 1 Detailed experimental parameters for the matrices used.**

| Sample matrix | SSD (mm) | Flow rate (L/min) | Hologram size (pixels) | Camera pixel size (μm) |
|---|---|---|---|---|
| PSL (air)[a] | 5 | 1.7, 0.7 | 2048 × 2048 | 5.5 |
| Snow meltwater (air) | 5 | 1.7, 0.7 | 2048 × 2048 | 5.5 |
| Ambient air | 5 | 1.7, 0.7 | 2048 × 2048 | 5.5 |
| Snow meltwater | 5 | Stationary | 2048 × 2048 | 5.5 |
| PSL (liquid)[b] | 5 | Stationary | 2048 × 2048 | 5.5 |
| 100-Fold PSL (liquid)[c] | 5 | Stationary | 2048 × 2048 | 5.5 |
| PSL + iron oxide (liquid) | 5 | Stationary | 2048 × 2048 | 5.5 |
| Glycerin + oil (liquid)[d] | 5 | Stationary | 2048 × 2048 | 5.5 |
| Zinc oxide (powder) | 5 | Stationary | 2048 × 2048 | 5.5 |
| Iron oxide (powder) | 5 | Stationary | 2048 × 2048 | 5.5 |

[a]Both 100 nm and 200 nm PSL spheres aerosolized (see 'Methods' section) and analysed by Nano-DIHM.
[b]The original colloid solution of PSL spheres (100 nm and 200 nm).
[c]100 μL of the original colloid solution of the PSL spheres mixed with 10 mL of Milli-Q water.
[d]100 μl of glycerin in 1 ml of type F microscope oil.

**Table 2 Three-dimensional size distribution of 100 nm PSL spheres in the aerosol phase.**

| Statistics | Width (μm) | Height (μm) | Length (μm) |
|---|---|---|---|
| Mean values | 0.264 | 0.265 | 0.739 |
| SD | 0.287 | 0.274 | 0.672 |
| Median values | 0.177 | 0.174 | 0.643 |
| 99th Percentile | 1.362 | 1.316 | 3.347 |
| 1st Percentile | 0.014 | 0.013 | 0.016 |

Descriptive statistics for the dimension distributions of the PSL spheres in 3D space in a single hologram.

transform to achieve the high resolution[51,76,77], and (3) several holograms are formed in an aqueous medium (water), and thus the wavelength of the laser is reduced[50,78] from $\lambda = 405$ nm to ~300 nm, which creates favourable conditions to achieve the high resolution. More details are provided in the 'Methods' section. We thereby successfully demonstrated the first-time in situ real-time observation of airborne nanosized objects using the Nano-DIHM (Figs. 1–4) and validated the results using aerosol sizers.

The Nano-DIHM effectively resolved the size and shape of airborne nanosized PSL particles with dimensions ranging from nano- to microscale. Figure 1a–n, Fig. 2a–c, Supplementary Fig. S1, and Table 2 provide information on the aerosolized 100 nm PSL particles, and Fig. 3a–o and Supplementary Fig. S2a–f provide information on the 200 nm aerosolized PSL particles. The PSL particle size and spherical shape determined by the Nano-DIHM matched the standard PSL particles' specifications (standard diameter, $101 \pm 7$ nm) for 100 and 200 nm PSL (standard diameter, $188 \pm 4$ nm). The size data also aligned with the results of simultaneous analysis of the aerosol size distribution in the outflow from the Nano-DIHM optical cuvette performed using the SMPS and the OPS instruments, measuring the aerosol size distribution in the particle size range from 10 nm to 10 μm (Fig. 1m, n).

Examples of intensity and phase reconstructions for the 100 nm PSL spheres in the moving air are depicted in Fig. 1a–h. The raw hologram in Fig. 1a was recorded for the airborne PSL spheres, which carried the information of phase, size, and shape, while the background hologram (Fig. 1b) was recorded without PSL particles and only dry air purified used with three high-efficiency particulate air (HEPA) filters, leading to a particle-free spectrum, and the contrast hologram (Fig. 1c) is a result of subtraction of the background hologram from the raw hologram. Subtracting the background from the raw hologram removes the

possible contamination due to the source (pinhole imperfections) and the object holder (a cuvette or microscope slide). Thus, this process significantly improves the reconstruction of an image, as discussed in numerous publications[50,58,79]. The reconstruction of the cropped area (c1) in Fig. 1c, as displayed in Fig. 1d, is of interest. Figure 1e is an example of high-resolution images of Fig. 1d; this high resolution is achieved by performing deconvolution and in-focus reconstruction, thus enhancing image quality, and reducing noise[50,58,69,74,76,80–82]. Figure 1f is a zoomed-in area of Fig. 1e, revealing information on nanosized objects and their shape.

Phase reconstruction of the identical hologram exemplified in Fig. 1g, h yields the replicate size and shape of airborne PSL particles as intensity images (Fig. 1e). The intensity profiles along the crosscut of particles (1 and 2) are shown in Fig. 1k, l, whereas their phase response is shown in Fig. 1i, j, respectively. The particle size determined by the Nano-DIHM along the crosscut through particle 1 is 160 and 200 nm, and 119 nm through particle 2 (Fig. 1f, h), expressed as the full width at half maximum (Fig. 1i–l). As shown in Fig. 1I, k, the Nano-DIHM successfully recovered the particle images within a lateral dimension of ~200 nm in agreement with the previous study[74,76], which stated the lateral resolution is achievable on the scale of a half of the wavelength ($\lambda/2$). The diameters of the PSL spheres determined by the Nano-DIHM vary from nano- to submicrometer-scale and are validated by the SMPS and the OPS data for the same samples (Fig. 1m, n). The 100 nm airborne PSL particle size determined by the Nano-DIHM agreed well with the particle size simultaneously measured using the SMPS and the OPS. During the numerical reconstruction process, many particles can be in focus at a given Z-value. Previous literature has shown similar results at a given Z-value and many particles were detected[43,50,58,60,65,78,79]. For example, Xu et al.[58] in Fig. 6 showed that several particles could be detected at a particular Z-value.

Figure 2a, b displays the 3D orientation/position of particles and the individual dimensions (width, height, and length) of PSL aerosol particles. The descriptive statistics of 100 nm PSL spheres in 3D space are provided in Table 2. The median values of the width, height and length determined by the Nano-DIHM are 177, 174, and 672 nm, respectively. The first percentile of the observed particle dimensions is 14 nm (width), 13 nm (height), and 16 nm (length), which is evidence that Nano-DIHM can be used for nanoparticle measurements. Furthermore, the SMPS size distribution of airborne PSL particles exhibits a good agreement with the width and height of particles observed by the Nano-DIHM and suggests that the most abundant PSL particle size peaks between 100 and 160 nm.

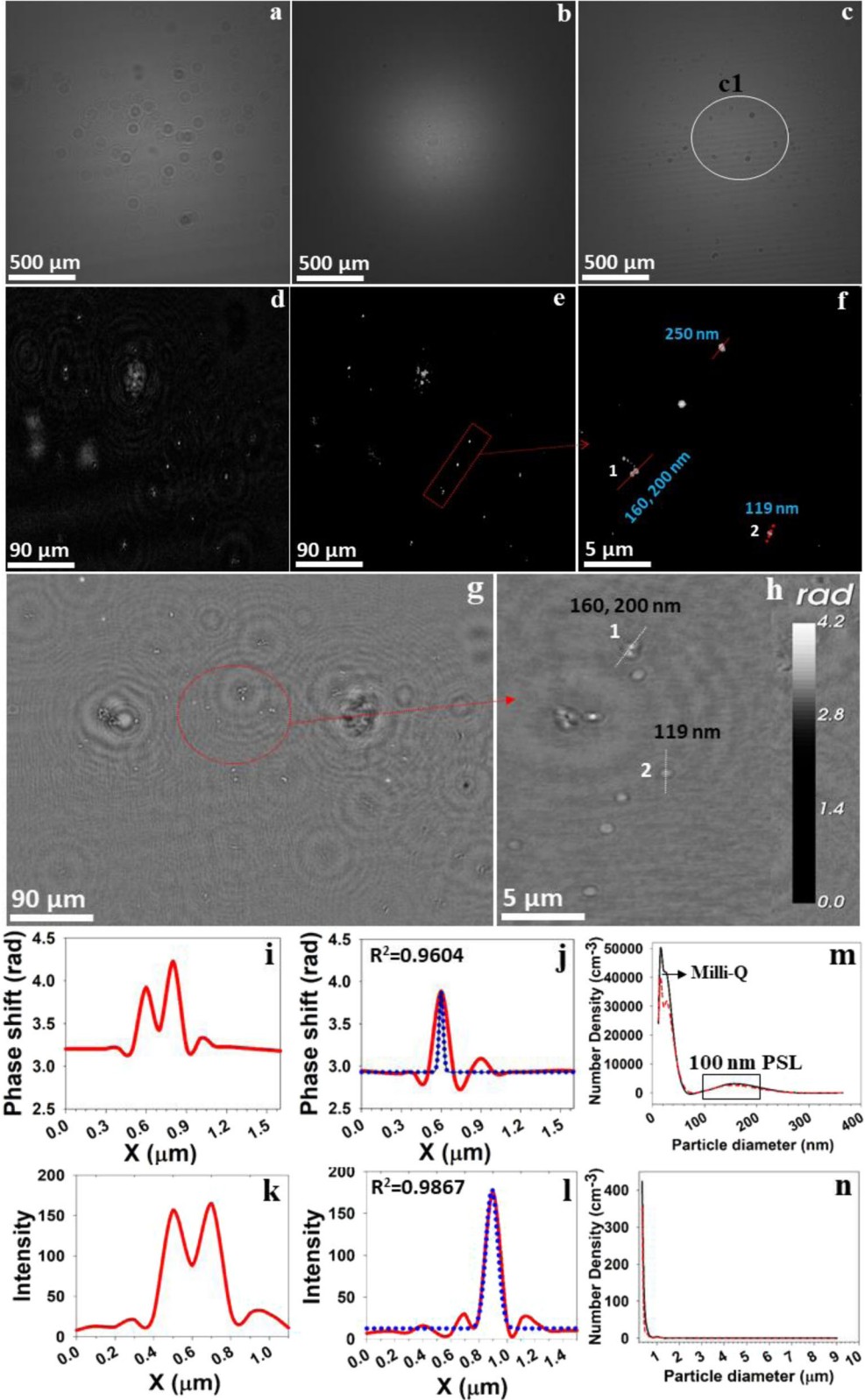

Another confirmation for positive observation of 200 nm airborne PSL particles measured in real time in situ by Nano-DIHM is shown in Fig. 3. The intensity and phase reconstructions of airborne PSL particles are displayed in Fig. 3a–i and their intensity responses across the crosscut of particles are shown in Fig. 3j–m. Nano-DIHM reveals the ~200 nm particles and particles of larger diameter ~1.4 μm (Fig. 3e). In addition, intensity and phase profiles through the crosscut of the PSL particles were determined for the ~400 nm, ~500 nm, and ~2 μm particles (Supplementary Fig. S4a–d). Nano-DIHM and the SMPS/OPS confirmed that the PSL spheres include a wide range of particle sizes ranging from 10 nm to 10 μm in the airborne state. The recovery of the shape and size information for a single PSL particle (Fig. 3c, f) unveils that Nano-DIHM can also be used for

**Fig. 1 Reconstruction of the intensity information for 100 nm PSL particles in the aerosol phase. a** Raw hologram recorded for airborne PSL particles; **b** background hologram recorded without PSL particles; **c** contrast hologram obtained after subtracting the background hologram from the raw hologram. **d** Zoomed-in area of c1 at $z = 3109\ \mu m$. **e** Particles in focus from **d**. **f** Zoomed-in area of **e** showing the precise recovery of nanosized PSL particles and their shape. **g**, **h** Phase reconstruction of PSL particles. The line across the numbers 1 and 2 in **f** and **h** shows the crosscut across the particles. **i**, **j** The phase profile of the PSL particle crosscut. **k**, **l** The intensity profile of PSL particles across the particle crosscut. **m**, **n** The size distribution of airborne PSL particle (aerosols) measured by the SMPS and the OPS, respectively. The red dashed line and the black line correspond to the two repeats of the experiments. The Blue dotted line is a Gaussian peak fit for the data. The validation of the holographic reconstruction of 100 nm PSL particles was compared with S/TEM images in Fig. 4. The 4D dynamic trajectories of the particles are provided in Supplementary Movies 1 and Supplementary Movies 2. The background holograms recorded for zero air and particle concentration tested by the SMPS and the OPS are shown in Supplementary Fig. S3.

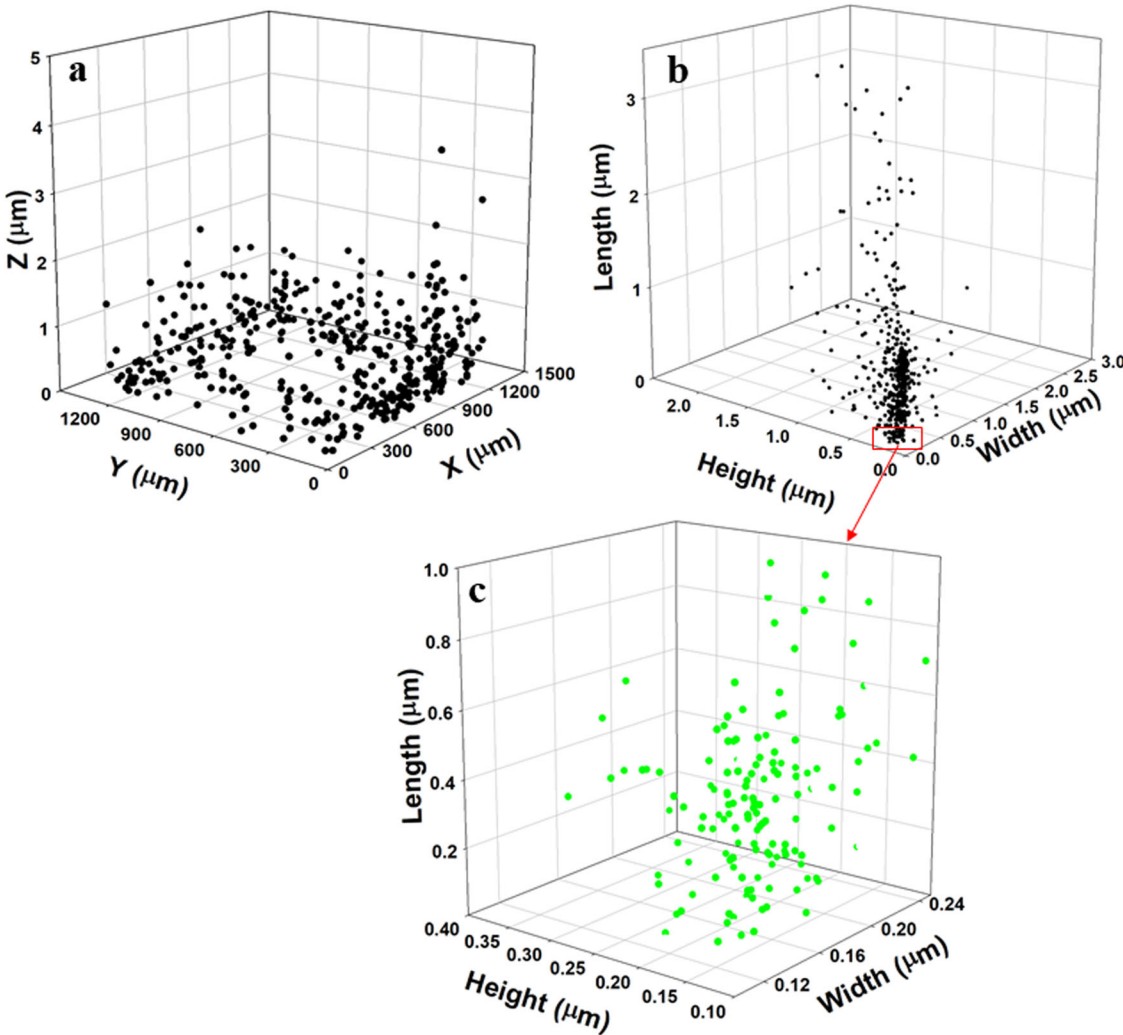

**Fig. 2 3D size distribution of 100 nm PSL spheres in the aerosol phase. a** The orientation of PSL particles. **b** Width, height, and length distribution of PSL particles in a single hologram over 62.5 ms. **c** Highlighted PSL particle size distribution from **b** within the nanosized width of particles. Another 3D reconstruction of different holograms is shown in Supplementary Fig. S1.

single-particle detection. The Nano-DIHM worked effectively not only for single aerosol particle detection but also allowed the multiparticle scanning of the total aerosol sample (Fig. 2a–c and Supplementary Fig. S2a–f).

The particle's orientation and dimensions are shown in Supplementary Fig. S2a–f and the tabular form of Supplementary Fig. S2c, f provides the statistics for the 200 nm PSL spheres in 3D space. The median values of the width, height, and length determined by the Nano-DIHM were 319, 319, and 678 nm, respectively. The first percentile of the observed particle dimensions was 23 nm (width), 23 nm (height), and 15 nm (length). The length of the PSL spheres was observed to equal almost twice the

width and height. It may be because two or more particles in different planes were closer than the depth resolution (362 nm); the Nano-DIHM could not distinguish them and observed them as a single particle. This limitation should be considered in future studies. We recommend the usage of further length correction methods along other routes to reduce or eliminate this limitation.

As seen, in both cases, the particle size distributions of the 100 nm and the 200 nm PSL spheres in the airborne state obtained by the Nano-DIHM agreed with the particle size distributions measured by the SMPS and the OPS (Figs. 1m, n and 3n, o). The observed size distribution of airborne PSL spheres

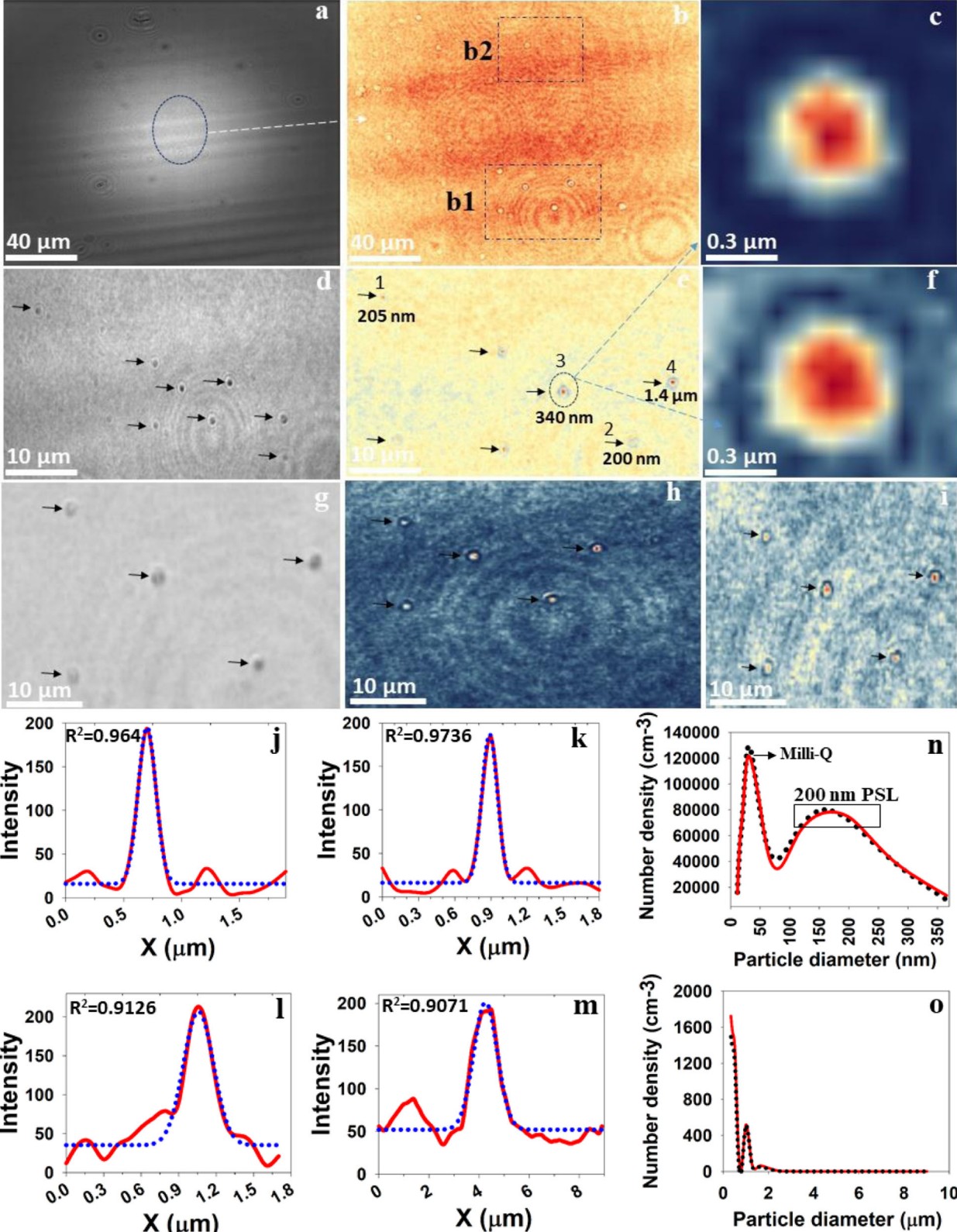

**Fig. 3 Holographic reconstruction of 200 nm PSL airborne particles. a** Raw hologram; **b** zoomed-in region of **a** at $Z = 1547\,\mu m$. **d**, **e** Phase and intensity reconstruction of the region of box b1 in **b**. The arrow mark indicating the airborne PSL particles from nanosized to the larger diameter of particles. **g–i** Phase, intensity, and amplitude reconstruction of PSL particles in the region of box b2 in **b**, respectively. **c**, **f** Zoomed-in intensity and amplitude images of the PSL particle clearly showing the spherical shape. **j–m** Intensity crosscut of particles 1, 2, 3, and 4 levels in **e**, respectively. **n**, **o** The airborne PSL particle size distribution confirmation by the SMPS and the OPS. Various other S/TEM and their holographic reconstruction comparisons are included in Fig. 5. More intensity crosscuts of particles are shown in Supplementary Fig. S4.

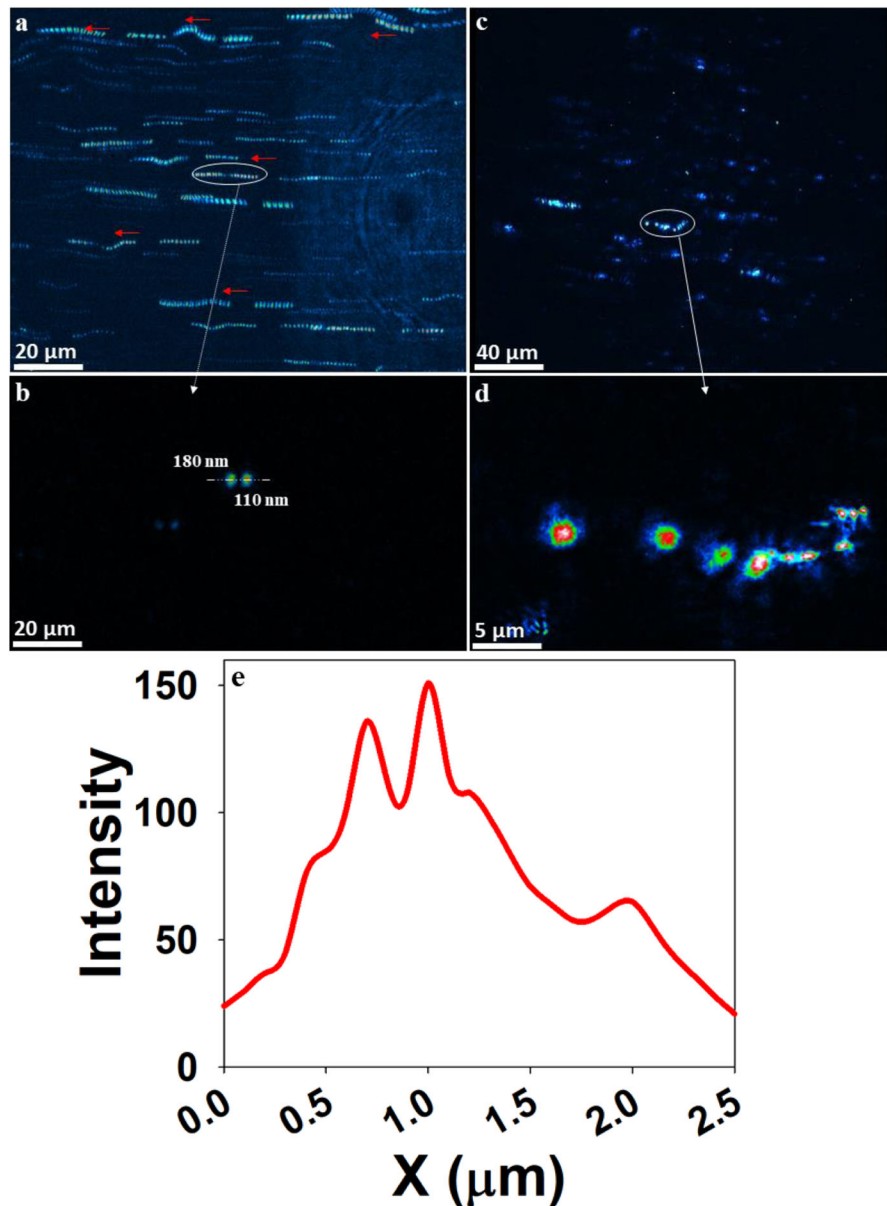

**Fig. 4 Trajectories of 100 nm airborne PSL particles in motion within the flow tube cuvette. a** Sum of 16 holograms with 62.5 ms temporal resolution. **b**, **e** The intensity response of selected particles and their diameters are 180 and 110 nm (FWHM), respectively. **c** Sum of nine holograms with 62.5 ms temporal resolution. **d** Reconstruction of summed hologram (**c**) in one plane with the trajectory of circled data in focus. The red arrows indicate the directions of particles motion.

obtained by both the Nano-DIHM and the SMPS and OPS was bimodal. This size distribution had a broader peak between 100 and 160 nm for the 100 nm PSL particles (Fig. 1m), and between 150 and 300 nm for the 200 nm PSL (Fig. 2n) corresponding to the 150 and 300 nm, consistent with the particles' median dimensions determined by the Nano-DIHM (Fig. 2c and Supplementary Figs. S1c and S2c, f).

The aerosolized Milli-Q water peak was observed by the SMPS and the OPS between 30 and 50 nm (Fig. S5). At the same time, the following median values for the same water sample were observed by the Nano-DIHM (Supplementary Table S1): 41 nm (width) and 43 nm (height). They aligned well with the SMPS and OPS measurements (Supplementary Fig. S5a–d). The Milli-Q water peak at around 30–50 nm is typical of various aqueous dispersions and arises from condensed dissolved compounds in the solution[14,83] or may arise during aerosolization[14]. The median and mean particle dimensions varied within ±5% from one

hologram to the next during reconstructions of several holograms (Supplementary Figs. S1 and S2). This uncertainty likely arises during hologram reconstruction due to the input digital image data that includes noise during the recording process.

## 4D trajectories of airborne PSL spheres

The dynamic Four-dimensional (4D) trajectories (3D positions and 1D time) of 100 nm PSL spheres are provided in Fig. 4a–d and Supplementary Movies 1 and 2 with a temporal resolution of 62.5 ms. The Supplementary Movie 1 is a zoomed-in image of Supplementary Movie 2. To obtain high-resolution trajectories of the airborne nanoparticles in this study, we used the following procedure: (1) a sequence of holograms was recorded at 16 fps. (2) The background contaminations were eliminated by subtraction of consecutive holograms and (3) the resultant holograms were reconstructed at a given plane and summed to obtain

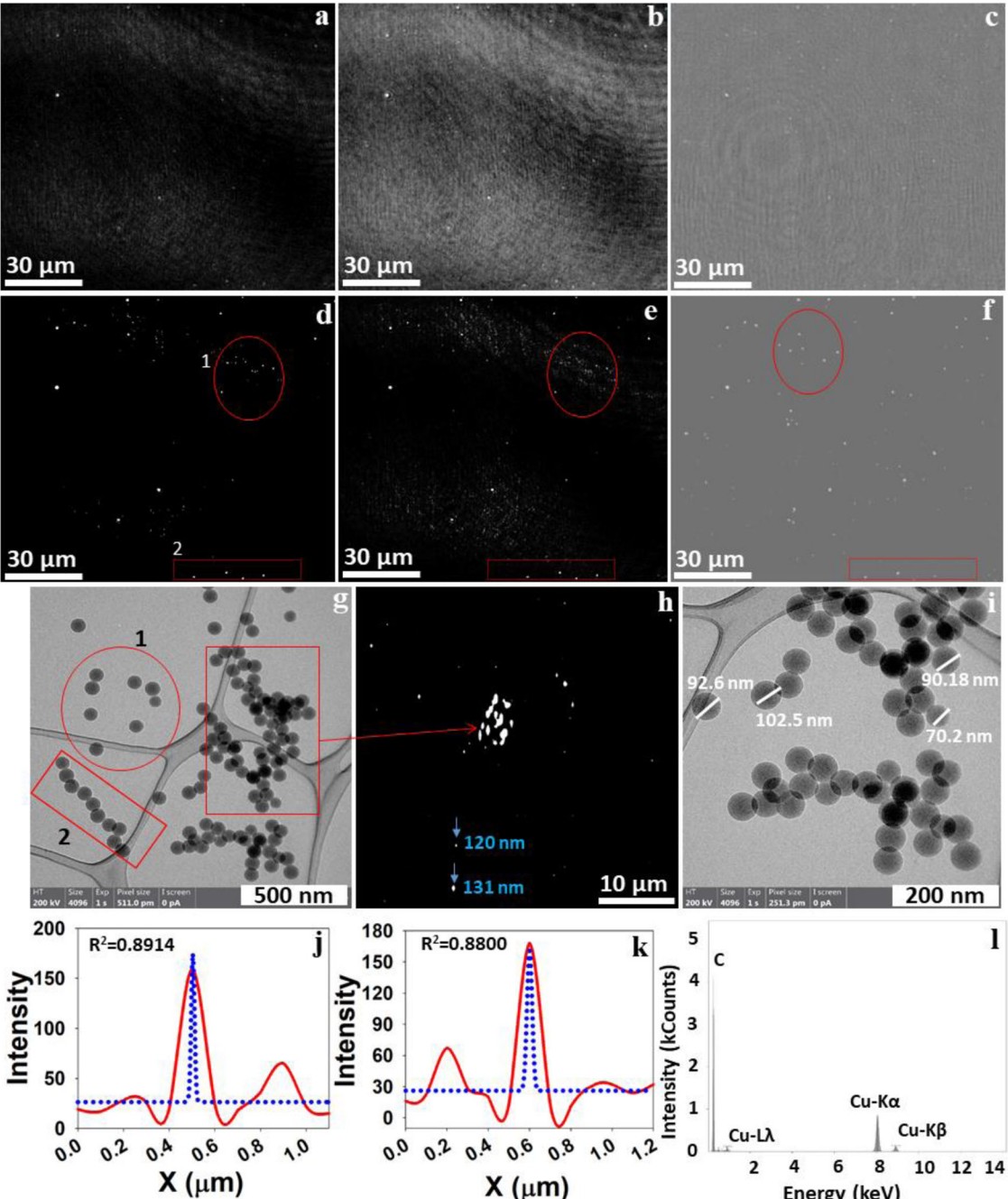

**Fig. 5 Holographic detection of 100 nm PSL particles deposited on a microscope slide. a–c** Intensity, amplitude, and phase reconstructions at reconstruction distance $Z = 1353\,\mu m$. **d–f** In-focus high-resolution reconstructions of images at $Z = 1347\,\mu m$ corresponding to **a–c**, respectively. **g, i** S/TEM images of the same 100 nm PSL samples. **h** Intensity reconstructions at $Z = 1650\,\mu m$. Circle 1 and rectangular box 1 in the image (**g**) are similar to the holographic reconstruction (**d–f**). **j–k** Intensity crosscut of particles shown in **h**. **l** EDS data of PSL particles. The Blue dotted curve in **j**, **k** presents the Gaussian peak fitting of the data.

the final trajectories (16 holograms summed in Fig. 4a and 9 holograms summed in Fig. 4c). Subtracting holograms ensured that the dynamic range was not exceeded and only the object-related information (moving PSL particles information) was retained[50,84]. Figure 4a, c shows the sequential positions at successive recording times of the airborne 100 nm PSL particles contained in the sample volume in two reconstructions planes separated by 200 μm. Figure 4b depicts the zoomed-in crop area of trajectories given in Fig. 4a to provide the confirmation of the size of 100 nm PSL airborne particles. Sixteen sequential positions clearly define the trajectory of airborne PSL particles (Fig. 4a),

which, moving in a random manner, confined to a plane parallel to the flow tube cuvette.

All the reconstructions of the holograms were performed at the same reconstruction distance ($z$) and then processed to create the Giff movies. As depicted in Supplementary Movie 1, the dark red particles are in the focus of the reconstruction plane, whereas some of the particles (green/blue) are slightly out of focus at the exact reconstruction distances due to the finite depth of field of the objective. As seen in Fig. 4a–d, some PSL particles are in the focus reconstruction plane, yet some are progressively out of focus, indicating that the motion direction also has a component

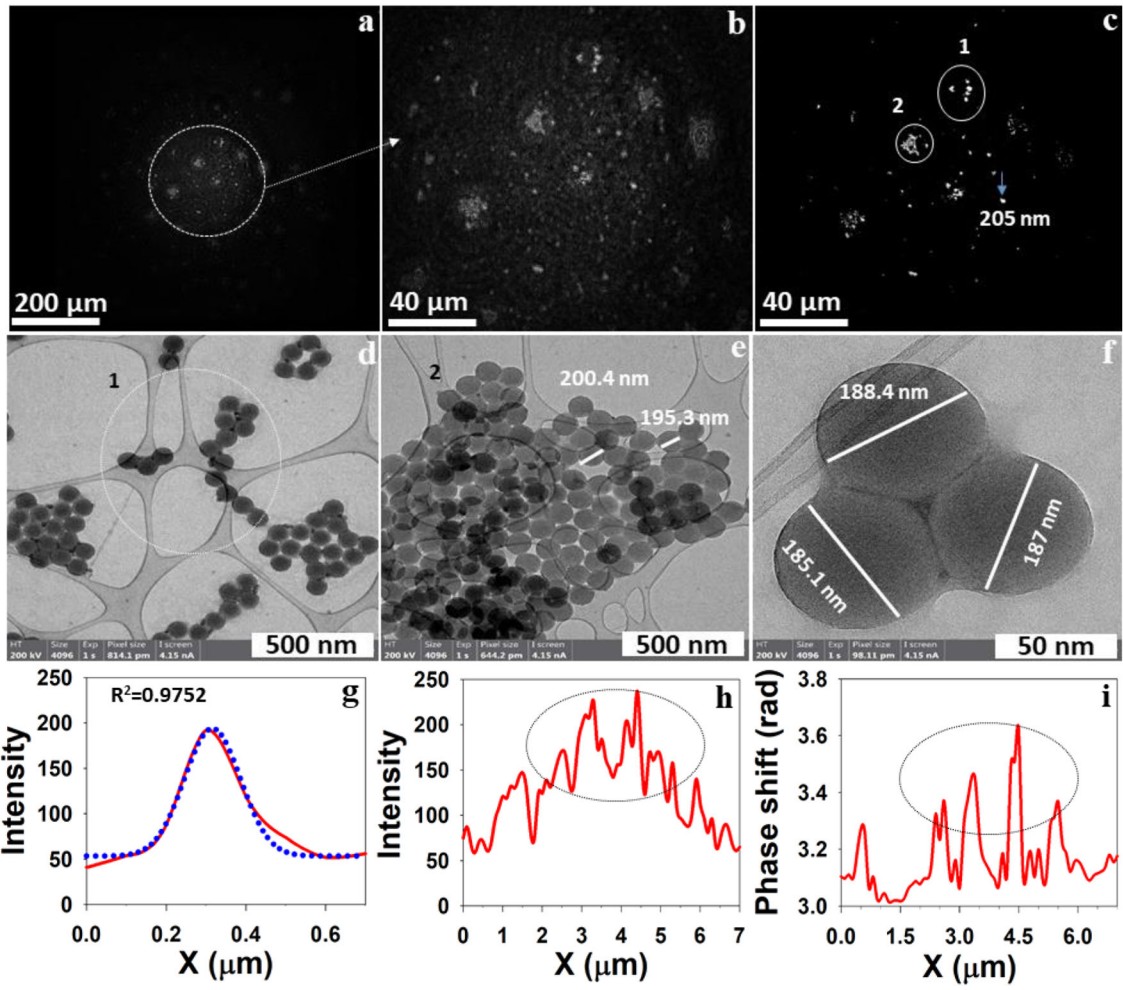

**Fig. 6 Holographic reconstruction of 200 nm PSL particles deposited on a microscope slide and their validation by S/TEM images. a** Intensity reconstruction at 2318 μm. **b** Zoomed-in area of **a**. **c** In-focus, high-resolution reconstruction at $Z = 2321$ μm. **d**–**f** S/TEM images of same 200 nm PSL samples. Circle 1 and 2 in image **c** show the good agreement with S/TEM images shown in **d, e. g** Intensity crosscut of particles in **c** with a diameter of 205 nm. **h, i** Intensity and phase crosscut of particles with the confirmation of existing clusters of PSL particles. Similar confirmation by S/TEM image in **e** suggested the presence of a cluster of PSL particles. Phase reconstruction of 200 nm PSL particles shown in Supplementary Fig. S6.

perpendicular to the reconstruction plane. To overcome the out-of-focus reconstruction in-complete trajectories analysis, several reconstructions from the same hologram in many planes are necessary[50,84]. As such, we demonstrated the ability of the Nano-DIHM to visualize particle trajectories.

**PSL spheres in aqueous mode: confirmation with S/TEM**
To evaluate whether Nano-DIHM can determine shapes, size, and morphology of individual PSL spheres in an aqueous state, we investigated the same samples of 100 nm PSL and 200 nm PSL particles using Nano-DIHM (Figs. 5a–f and 6a–c) and S/TEM (Figs. 5g, i and 6d–f). Figure 5a–f shows the reconstruction of the holograms in all three modes: intensity, amplitude, and phase. Figure 5d–f is a high-resolution image of Fig. 5a–c and illustrates the PSL size, shape, and phases. The red circles and rectangles marked in Fig. 5d–f highlight an example showing the consistency of the existence of 100 nm PSL particles in all three reconstruction modes and the alignment of these Nano-DIHM measurements with the particle size and shape determined by S/TEM (Fig. 5g, i).

As shown in the S/TEM images (Fig. 5g), the red circle 1 and rectangle 2 revealed the similarity to the PSL particle shape obtained by Nano-DIHM (Fig. 5d–f). The S/TEM image (Fig. 5i)

confirmed that the PSL particle size ~100 nm is in a good agreement with the observed median values of PSL particle width (110 nm) and height (120 nm) obtained in 3D space by Nano-DIHM (Supplementary Table S2). Thus, the observations of PSL particle size and shape made by S/TEM and Nano-DIHM are well-matched with the size of the manufactured PSL spheres (standard diameter $101 \pm 7$ nm). The intensity response of particles along the crosscut is shown in Fig. 5j, k. The energy-dispersive X-ray spectroscopy (EDS) data for PSL particles presented in Fig. 5l confirmed the PSL sample quality.

We also show another example of validation of particle size and their shape made by Nano-DIHM with S/TEM (Fig. 6). Here we examined a 200 nm PSL particle deposited on a microscope slide. The size and shape determined by Nano-DIHM (Fig. 6a–c) are complemented with S/TEM images (Fig. 6d–f). In Fig. 6c, circles 1 and 2 show PSL particles imaged by Nano-DIHM are parallel to images made by S/TEM shown in Fig. 6d, e. The intensity response along the crosscut of the PSL particle shown in Fig. 6g indicates a single particle with a diameter of ~198 nm, which is similar to the size provided by the manufacturer (standard diameter $188 \pm 7$ nm) and the size measured by the S/TEM (Fig. 6e, f). The S/TEM images (Fig. 6e) confirmed that PSL particles existed as agglomerates or clusters. The Nano-DIHM also confirmed the presence of agglomerates/clusters (Fig. 6h, i).

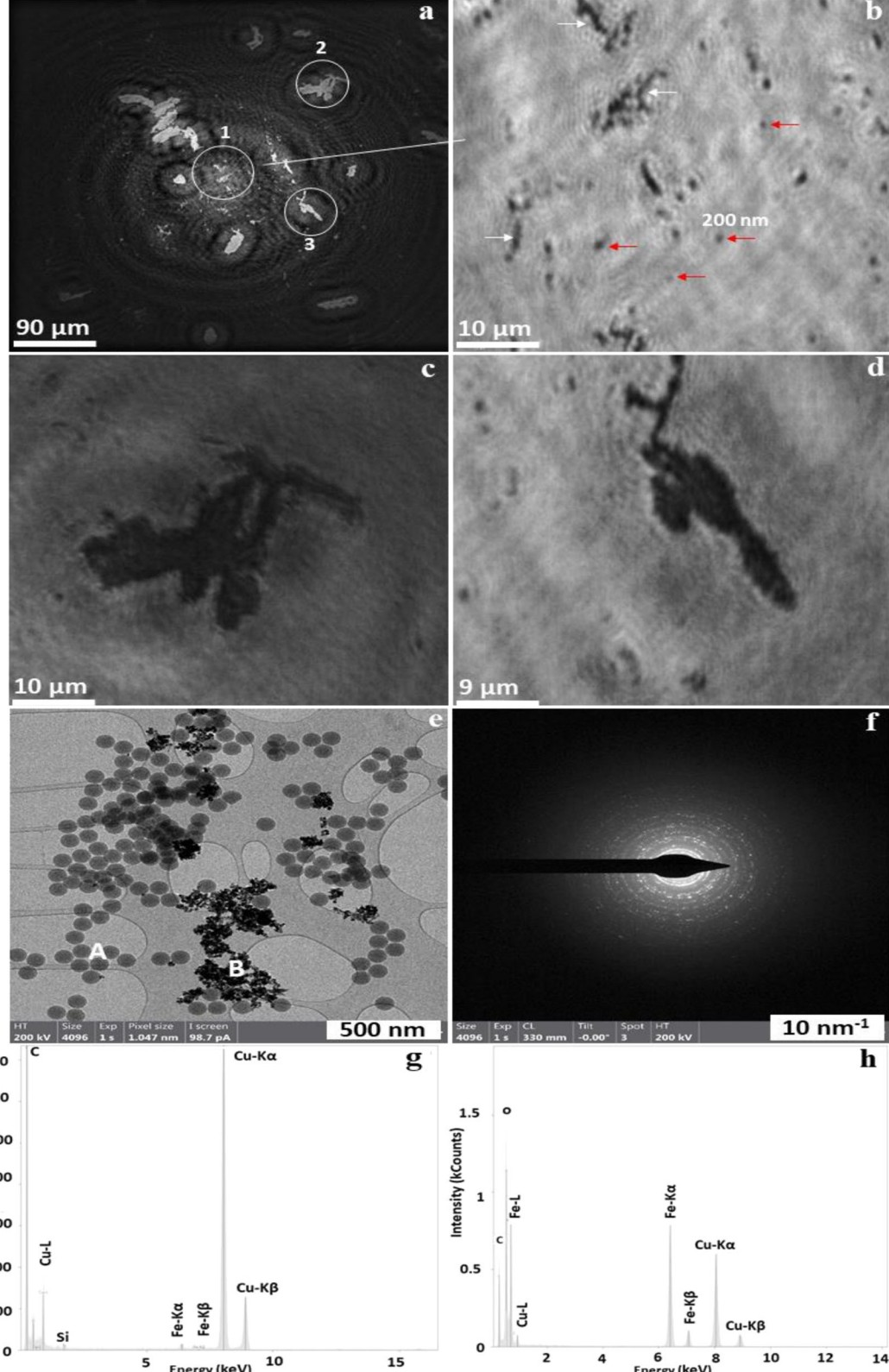

**Fig. 7 Holographic reconstruction of a mixed sample of 200 nm PSL particles and iron oxide nanoparticles deposited on a microscope slide and the validation by S/TEM images. a** Intensity reconstruction at 905 µm. **b** Zoomed-in area circle 1 of **a**. **c**, **d** Zoomed-in area circles 2 and 3 of **a**. **e** S/TEM images of the same sample confirming the presence of both PSL and iron oxide nanoparticles. Circles 1, 2, and 3 are shown in **a** and highlighted in **b**–**d**, respectively. **f** Selected area diffraction pattern and **g**, **h** EDS data for Spot A and Spot B, respectively. Phase reconstruction of the same samples and more S/TEM images are provided in the Supplementary Fig. S7.

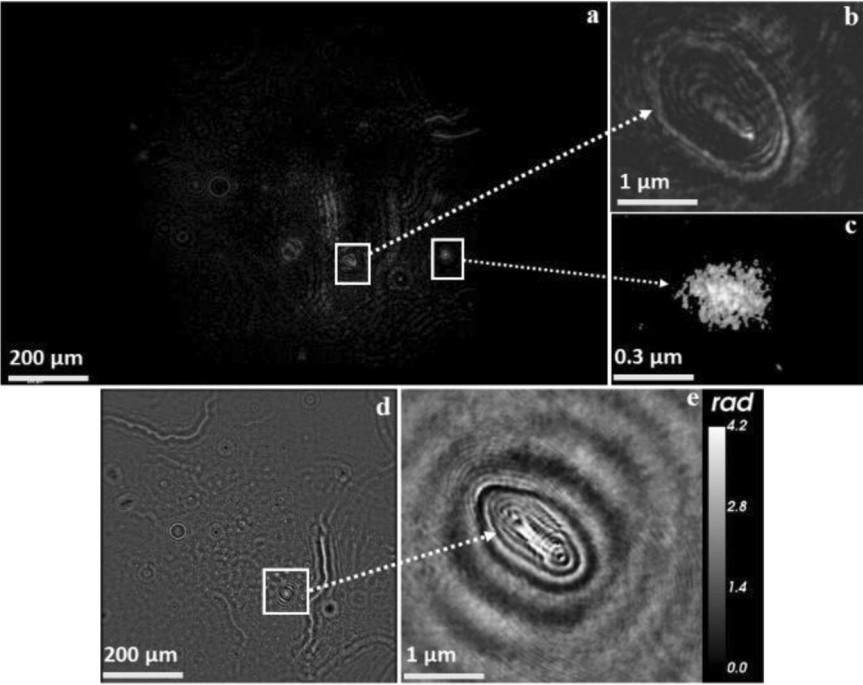

**Fig. 8 Hologram reconstruction for ambient air showing different airborne particles' sizes, shapes, and morphologies. a** Intensity reconstruction of airborne particles. **b**, **c** High-resolution intensity images. **d** Phase reconstruction of image **a**. **e** Phase image. The phase of particles through a cross-section were shown in Supplementary Fig. S8a–f. Simultaneously aerosol size distribution obtained from the SMPS and the OPS were shown in Supplementary Fig. S9a, b.

Although the Nano-DIHM identified the overall shape of particles, it did not decipher the precise cluster shape of particles as the S/TEM did. The Nano-DIHM thereby provides promising results to determine the phase, shape, and size of the particles, yet high-resolution electron microscopy is a valuable tool for determining the refined shape, at this stage, despite the lacking in situ and real-time imaging capabilities that Nano-DIHM offers.

### Distinguishing PSL spheres from iron oxide mixture: S/TEM

The shape, size, and morphology of PSL and iron oxide particles within the mixed sample of PSL particles and iron oxide particles have been successfully determined using Nano-DIHM. Figure 7a shows the reconstructed intensity image of mixed PSL and iron oxide particles on a microscope slide. The zoomed-in area in Fig. 7a is shown in Fig. 7b–d. As depicted in Fig. 7b, the red arrows indicate the PSL particles and the white arrows show the iron oxide particles or the iron oxide particles to which PSL particles are attached. The mixture of the same samples was also analysed using S/TEM. The S/TEM images (Fig. 7e) confirmed the size, shape, and morphology of the PSL, and iron oxide particles determined by Nano-DIHM (Fig. 7a–d). The EDS data for Spot A and Spot B (Fig. 7e) shown in Fig. 7g, h confirmed the identity of the PSL and iron oxide particles. The selected area electron diffraction pattern is shown in Fig. 7f. High-resolution images in Fig. 7c, d (zoomed-in areas 1 and 2 in Fig. 7a) depicted the different sizes and shapes of iron oxide particles and confirmed the attachment of the PSL particles.

The phase reconstruction of the same samples shown in the Supplementary information (Supplementary Fig. S7) complements the S/TEM images. As discussed in this example, Nano-DIHM showed the capability to determine the size, shape, and morphology of the synthetic materials within the mixed samples and successfully distinguish the nanosized particles.

### Tracking ambient airborne particles

Figure 8 illustrates examples of the capability of the real-time analysis of ambient airborne particles using Nano-DIHM. For instance, the intensity and phase reconstruction of real-time Nano-DIHM measurements of particles in ambient air is shown in Fig. 8a, d. Figure 8b, c shows two selected areas of Fig. 8a, identifying particles of two different sizes and shapes in the ambient air. The morphology of the particle in Fig. 8c confirmed the presence of agglomerates. It is noteworthy that although the intensity image in Fig. 8b does not indicate the particle's precise structure, the phase image of the same particle in Fig. 8e resolves the particle's structure and morphology clearly. The morphology of the particle in Fig. 8b, e suggests its biological origin. Particles with a similar morphology have been observed in previous studies using transmission electron microscopy[23,85], which supports this particle's organic or biological origin. Yet, further genomic confirmation will be required for the positive determination of airborne microbiological entities. We observed the changing phase shift of the particle in Fig. 8e across crosscuts and this is shown in Supplementary Fig. S8a–f. Quantitatively, changing phase shifts of 2.4 to 4.2 radians were observed across particle crosscuts (Supplementary Fig. S8c–f), suggesting the presence of multiple layers or particle heterogeneity.

### Discussion

This study demonstrates the first real-time acquisition of size, shape, phase, and morphology information for free-flowing spherical and non-spherical particles in moving air in situ using Nano-DIHM. The successful use of the developed technique was also demonstrated for non-gaseous environmental matrices: water samples from the ambient environment and solid synthetic materials (zinc oxide and iron oxide) (Supplementary Note 3).

Nano-DIHM efficiently resolves the size and shape of PSL particles in moving air, demonstrating the newly developed

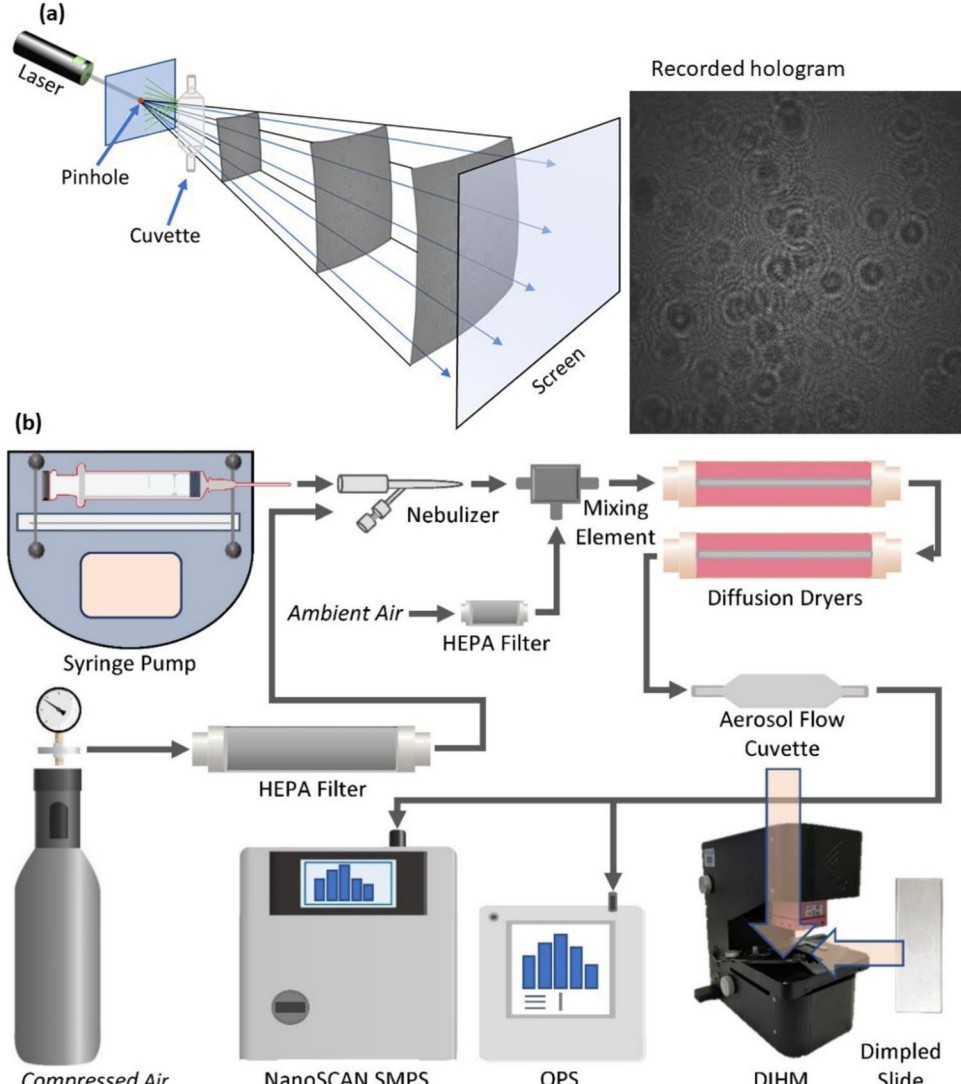

**Fig. 9 Digital in-line holography microscopy setup. a** Schematic showing the operational principle of DIHM and a recorded hologram. **b** Schematic of the experimental setup for holographic imaging in moving airflows, stationary liquid, and solid samples by Nano-DIHM.

technique's ability to describe aerosol dynamics with rapid sequential hologram acquisitions. The real-time dynamic trajectories of aerosols in atmospherically relevant conditions open the vast potential to investigate physicochemical properties of nanosized or micrometre-sized particles and their impact on climate and human health[57,86]. Nano-DIHM successfully observed the original spherical shape and size of airborne PSL particles ~119 nm (Fig. 1f) in 2D space, whereas in 3D space, it detected nanoparticles ~10 nm (first percentile; Table 2 and Supplementary Fig. S1c). Previous studies suggested that in holographic microscopy, in the volumetric reconstruction (3D space), particles/objects can be localized within 1 nm[87] in all three dimensions. Besides the detection of smaller particles, Nano-DIHM also determined the larger diameter particles (Fig. 3 and Supplementary Fig. S4a–d). The size distribution of PSL particles in the aerosol phase obtained by Nano-DIHM was validated by the SMPS and the OPS techniques (Figs. 1m, n and 3n, o).

In the ambient air, our results indicate that the Nano-DIHM technique can distinguish a broad range of particle sizes (nano-sized to submicron-sized), resolving spherical, irregularly shaped particles and particles of varying morphology. The measurement of the real-time phase shift of ambient air particles was also

achieved (Supplementary Fig. S8a–f). The ability to accurately measure the real-time phase shift and the refractive index is significant in the characterization of ambient air particles. Furthermore, it allows distinguishing between particles of different compositions, liquid vs. solid particles, such as water droplets and ice crystals in real time and in situ. For example, the analysis of the size and phase of snow-borne particles in the dynamic and stationary mode has been discussed in the Supplementary Note 1. Nano-DIHM successfully recovered the phase and size of snow-borne particles (Supplementary Fig. S12) and S/TEM validated their morphology (Supplementary Fig. S13).

The Nano-DIHM accurately determined the refractive indices of moving aerosols in the gas phase. We determined the refractive indices of particles in a liquid suspension (Supplementary Note 2). The reconstructions at different crosscuts through particles validate that Nano-DIHM effectively distinguishes refractive index variations of 0.001. Such results are consistent with the past studies[78,88].

Prior to this study, only optical trapping techniques enabled refractive index measurements for aerosols. Accurate determination of the refractive index allows inferences about particle composition[89,90]. For instance, optical chromatography has been previously used to identify the chemical composition of particles

based on their size and refractive index[89–91]. The combination of theoretical and experimentally measured refractive index values has also been used to distinguish particle types[90,92]. The considerable improvement of the resolution and the lower size limit of particles amenable to analysis by Nano-DIHM opens a range of new applications for the technique in future by permitting real-time and in situ measurements of the phase, size, shape, and refractive indices of particles.

The attained temporal resolution of 62.5 ms can still be improved to the level of microseconds through further optimization, such as changing the frame rate of image acquisition, the shutter speed, or using a detector with a higher resolution[65,70,78]. Furthermore, the successful implementation of fully automated Stingray software with Nano-DIHM would allow the reconstruction of thousands of holograms within an hour's interval time, allowing the exploration of the aerosol's complex physical and chemical processes in the future[70]. We have started to train the Stingray software[75], and some preliminary results of the airborne PSL spheres are shown in Supplementary Fig. S10. However, significant further progress is required. The ability of Nano-DIHM to visualize the motion of nanoscale and microscale particles in 3D space opens opportunities in multiple facets of fundamental and applied aerosol science. The capability of Nano-DIHM to probe aerosols in situ will allow exploring physicochemical processes under atmospherically relevant conditions.

The potential of Nano-DIHM in the atmospheric sciences is in the context of investigating the dynamics of phase transitions: particle freezing processes,[19,93] and efflorescence and deliquescence[69,94] in fine aerosol particles, which are poorly understood processes critical to unravelling the effects of aerosols on climate[4,95]. Recent evidence shows that phase transitions in atmospheric particles are more complex than predicted earlier, and these processes can happen through multiple unknown pathways[8,94,96]. We expect that the direct imaging of fast temporal changes in the morphology of particles with Nano-DIHM will prove invaluable for understanding the complex mechanisms of ice nucleation phenomena. Furthermore, the broad applicability of this new experimental technique is expected to open new directions in applied and fundamental particle research. Next-generation software and more advanced hardware will likely advance the performance of Nano-DIHM for remote and automated observation of complex matrices, such as planetary atmospheres.

## Methods

**Digital in-line holographic microscopy**. DIHM is a two-stage process. At the first stage, holograms are recorded. At the second stage, the reconstruction of holograms is performed, yielding a visualization of the object(s). With the current setup, holograms were recorded using the 4Deep Desktop Holographic Microscope[97]. The numerical reconstruction of the holograms was performed using the Octopus software package, version 2.2.2[52].

**Recording of holograms**. The schematic of the DIHM is shown in Fig. 9a. A laser (L) emits at the wavelength of $\lambda = 405$ nm. The laser beam is focused on the pinhole (P) that has an aperture diameter approximately matching the wavelength of the laser source. The resulting wave illuminates object(s) under observation from a distance of a few micrometres from the pinhole, producing a highly magnified diffraction pattern, the hologram, on the screen (photosensitive matrix, model MV1-D2048-96-G2-10, Photonfocus AG 00140622, version 2.1)[50,58]. We used the complementary metal oxide semiconductor (CMOS) photosensitive matrix screen that allowed recording holograms that are 2048 × 2048 pixels (5.5 μm camera pixels), stored as digital images for subsequent reconstruction[52,97].

As shown in Fig. 9a, the light emitted from the point source (the pinhole) propagates towards the screen and is scattered by the objects in its path, resulting in a hologram. The wave amplitude of the hologram at the screen, $A(r, t)$, is given by Eq. 1.

$$A(r, t) = A_{\text{ref}}(r, t) + A_{\text{scat}}(r, t) \tag{1}$$

where $A_{\text{ref}}(r, t)$ and $A_{\text{scat}}(r, t)$ are the reference and scattered amplitudes, respectively. The intensity of the hologram recorded on the photosensitive matrix screen, $I(r, t)$, is

$A(r, t)A^*(r, t)$ and can be expanded to yield

$$I(r, t) = A_{\text{ref}}(r, t)A_{\text{ref}}^*(r, t) + \left[ A_{\text{ref}}(r, t)A_{\text{scat}}^*(r, t) + A_{\text{scat}}(r, t)A_{\text{ref}}^*(r, t) \right] + A_{\text{scat}}(r, t)A_{\text{scat}}^*(r, t) \tag{2}$$

In Eq. 2, the first term is the intensity of the beam in the absence of a scatterer and the last term is the intensity of the scattered wave. The second term in the square brackets in Eq. 2 signifies the interferences between the reference and the scattered waves, referred to as holograms.

$$A_{\text{scat}}(r) = \frac{iA_{\text{ref}}}{r\lambda} \int \int I(r) \frac{\exp\left(ik\frac{rr'}{r}\right)}{|r - r'|} ds \tag{3}$$

The distance between the source (pinhole) and the camera sensor (screen) is ~5 mm, the wavelength of light is 405 nm and the camera pixel size 5.5 μm. These are the input parameters for the reconstruction. No other information is required to retrieve images, which yield the position, orientation, and shape of the observed objects. The 3D morphology and the 3D position of the object(s) can be determined from the reconstructions at different image planes. The proper reconstruction position is essential to obtain sharp images (Supplementary Fig. S11). Supplementary Fig. S11 shows that merely small changes of reconstruction position Z can significantly enhance image quality, including its resolutions. The aim was not to focus on the dimension of the particles. The reconstruction plane is defined as the plane between the laser source's focus and the camera sensor parallel to the YZ- plane[50,57,98]. These distances are found manually by systematically changing the reconstruction positions, i.e., the distance between the focus of the laser and the reconstructed plane[50,57]. The absolute dimensions of the observed object(s) in the reconstructed image are calculated during the resultant hologram's inversion. No further calibration is needed for the determination of the dimensions of the observed object(s). We show the quality of background holograms with our experiment in Supplementary Fig. S3a. The mean values of background intensity along the different crosscuts of holograms are 8.05 ± 6.56 arbitrary units (Supplementary Fig. S3a). The background hologram was recorded with zero air. The air or aerosol exiting the gas cuvette is passed into the SMPS and the OPS. The data obtained with the SMPS and the OPS for zero air showed a particle count of fewer than 1 particle/cm³ (Supplementary Fig. S3b, c).

**Resolution of Nano-DIHM: beyond the diffraction limit**. The resolution of reconstructed DIHM images depends on the information recorded in the hologram. In turn, the information captured in the hologram depends on numerous factors, such as (1) the pinhole size, controlling spatial coherence and illumination cone; (2) the numerical aperture (NA), given by the size and positioning of the recording photosensitive matrices, i.e., a charge-coupled device (CCD) or a CMOS chip; (3) the pixel density and the dynamic range, controlling fringe resolution and the noise level in the hologram; and (4) the wavelength[50].

The lateral resolution is represented in terms of the geometric parameters of DIHM. The recording screen is perpendicular to the optical axis connecting the light source with the centre of the screen. The lateral resolution of DIHM can be calculated using Eq. 2. A detailed description of the lateral criteria of DIHM has been reported in previous studies[50,58]. The holographic term in Eq. (2) can be rewritten as

$$|r2 - r1| \geq \frac{\lambda}{2\text{NA}} \tag{4}$$

Where NA is the numerical aperture of the microscope, and it can be defined as:

$$\text{NA} = \frac{W}{2\left[ \sqrt[2]{\left(\frac{d}{2}\right)^2 + (d)^2} \right]} \tag{5}$$

where $W$ is the width of the screen and $d$ is the distance between the point source and the screen. Depth resolution in holographic reconstruction is harder to achieve which improves with the increasing size of the screen[50]. The depth resolution of the holographic microscope is given by Eq. 6.

$$|r2 - r1| \geq \frac{\lambda}{2(\text{NA})^2} \tag{6}$$

The theoretical lateral resolution calculated for the setup used in this study was 271 nm, while the depth resolution was 362 nm. However, applying the experimental approach and the numerical process can bring a much lower resolution. In this study, these are the three processes used to overcome the diffraction barrier of the setup and detect the nanosized objects.

*Improved experimental approach*. In this study, we do not need to add any additional physical objects and there is no need for external filtering for the recording/reconstruction of holograms[50,52,75,79]. The holograms were recorded near the pinhole and bring the camera closer to the quartz flow tube cuvette for dynamic mode or quartz microscope slide for stationary mode. This configuration is possible, because the Octopus software[52] still allows us to record the hologram in the size of 2048 × 2048-pixel with a pixel size of 5.5 μm given the source-to-camera distance (5 mm). Hence, we achieved a larger field of view, allowing tracking both single-particle and multiple particles. The pinhole and the flow tube (cuvette) for

dynamic experiments or microscopy slides for the stationary experiment nearly touched each other. In DIHM, the shorter the source-to-sample distance, the higher the magnification and hence a higher resolution can be achieved[50,79,99]. By using this process, the NA of the Nano-DIHM increases substantially (NA = 0.7428) (Eq. 3). Thus, following the steps below, the nano-DIHM resolution can go below this theoretical threshold[50,76,78,79,81,99] and allow nano-size particle detection.

Keeping the cuvette near the source and bringing the camera to near the cuvette collectively increases the Nano-DIHM magnification[50,79]. Previously, the only way to increase the resolution was by numerical approaches, because the low NA was one of the major issues in achieving the high resolution in digital holography[50,76,78]. Another advantage of the current experimental setup shown in Fig. 9a is that the single pinhole can work with multiple virtual illumination sources. Thereby, we do not need multiple illumination sources to record the hologram. To obtain such short distances, the sample flow tube is introduced between the pinhole and CMOS so that the sample is facing the pinhole, as illustrated in Fig. 9a, b. The pinhole emits the light, and the bottom of the surface of the sample carrier will partially reflect this light to the pinhole. In turn, this light is reflected to the sample carrier, where it is partially reflected backward, multiple times. As a result, the light coming directly from the pinhole is superposed upon the reflected light, which appears to come from several virtual pinholes[99]. This experimental recording process of holograms enabled us to overcome the diffraction barrier, allowing measurement of the nanosized particles ≤200 nm. It is noteworthy that several studies using near-field optical microscopy have successfully captured the nanoscale objects, overcoming the diffraction limit of the optical system[31–35,37,42].

In digital holography, recent studies achieved detection of nanoparticles using an on-chip microscope where each nanoparticle served as a nucleus for self-assembled deposition of refractive materials (polyethylene glycol-based solution) around each particle (nanolenses), thus increasing each particle's size and scattering cross-section, effectively helping it's the detection on a chip[43,44,60,72,100,101]. Several theoretical numerical methods based on the deconvolution algorithm on the sensor chip and point source are used to detect the nanoparticles[45,87,102–104].

As the distance between the source and the sample gets shorter, the object vibrations and noise are getting more extensive due to the higher magnification[74,80]. It blurs out the fine interference fringes and reduces the potentially achievable high resolution. Indeed, it is a challenge and we agree with it. However, experimentally, to overcome this significant challenge and compensate for the vibrations and noise, a short-time acquisition sequence of the holograms and a thin sample carrier (compared to the distance between the pinhole and the image sensor) was used and recorded[80,105]. Such numerical reconstruction methods with deblurring techniques have been demonstrated that in the motion-deblurring photo-image analysis permits achieving a higher resolution by a factor of 2 or more[52,74,80].

*Numerical reconstruction process.* The Octopus/Stingray software[52,75] was used to reconstruct the recorded holograms in this study. The software is based on a patented algorithm[51], which can effortlessly achieve the lateral resolution on the order of half-wavelength ($\lambda/2$) of source and depth resolution on the order of one wavelength ($\lambda$)[76]. Furthermore, as shown in detail elsewhere[74,76,80,81], a higher resolution was achieved using multiple deconvolutions routines during the reconstruction: (1) illumination system (pinhole) and (2) the finite NA of the recording screen (CCD/CMOS)[74,76,80,81]. Implementing an instant 3D-deconvolution routine in our reconstruction method allowed us to reach the desire resolution[52,73,76,80,81]. The importance is that to find the plane where the phase/intensity image is accurately focused. Thereby, if we are aimed to measure the PSL for 200 nm size, as an example, the plane must be within 0.01 micron. Otherwise, the dots will be only a few pixels and do not look like quality images. For that reason, increasing the precision with 0.001 in Octopus software allowed us to achieve high resolution. Finally, we accurately focus on blurred objects by adjusting reconstruction position ($z$) to up to three decimal places (0.001).

The advantage of the deconvolution routines is to remove white noise from final reconstructed images and enhance the resolution by a factor of 2, as discussed in detail in previous studies[73,74,76,80]. By employing the 3D-deconvolution routines, the out-of-focus signal is brought back to its scatterer and the twin images are automatically removed, as they are not part of the scattered wave. Thus, spatially well-localized parts of an object are recovered free from artifacts and noise-free. For example, Nickerson et al.[76] in Fig. 5 shows how implementing the two-fold deconvolution mechanism removes the white noise from reconstructed images.

*Aqueous medium.* In addition to air, several holograms are formed in an aqueous medium (water) and thus the wavelength of the laser is reduced[50,78] from $\lambda = 405$ to ~300 nm.

**Phase shift and refractive index calculation**. The recorded holograms provided the information on the phase, size, and morphology of objects. The holograms were reconstructed using the Octopus software (version 2.2). The relative phase shift (rad) has been measured for the objects. Suppose the refractive index of objects is $n_{obj}$ and the refractive index of fluid or medium is $n_m$; the phase shift depends on

the distance that light travels in each medium. In this study, the light with wavelength $\lambda$ (405 nm) travelling a distance of $t$ in the object, results in the relative phase shift given by

$$\Delta\psi = \left(n_{obj} - n_m\right)\frac{2\pi t}{\lambda} \qquad (7)$$

Further, the optical path difference is:

$$OP = \left(n_{obj} - n_m\right)t = \frac{\lambda\Delta\psi}{2\pi} \qquad (8)$$

The crosscuts (phase shift vs. position of the object) through a phase reconstruction image of an object determine the optical path as a function of position within the objects. If the size of the particles/objects in the light propagation direction is known, the changing refractive index can be calculated from the measured phase shifts using Eq. (7). Otherwise, information on the refractive indices of the objects/particles and the medium allows determining the object/particle size in the light propagation direction (object height).

**SMPS and OPS**. A NanoScan™ SMPS model 3910 (TSI, Inc.) and an OPS model 3330 (TSI, Inc.) were used to measure the real-time size distributions of airborne particles. The SMPS measures the size of the airborne particles in the range of 10 to 400 nm in terms of the electrical mobility diameter. The sampling flow rate of the SMPS was 0.75 L/min. The OPS determines the particle size distribution in a size range of 0.3–10 μm in terms of optical diameter with a sampling flow rate of 1 L/min. A more detailed description of the SMPS and the OPS is provided elsewhere[15].

**Experimental setup**. The experimental setup (Fig. 9b) for measurement of airborne particles consists of the following components: the DIHM instrument, a gas flow cuvette (ES Quartz Glass, volume of 700 μL, path Length 2 mm), a microscope slide (Quartz Glass), the aerosol generator unit, and the aerosol sizers (SMPS and OPS). Each experiment has been triplicated (repeated thrice). During each repeat, we generally recorded 1000 holograms. The details of the aerosol generation unit and the particle sizers have been described elsewhere[15,20].

Several sample matrices have been tested (Table 1): ambient air, snow meltwater, aerosolized 100 nm and 200 nm size PSL, a mixture of PSL with iron oxide powder, synthetic materials such as zinc oxide (powder, <60 nm particle size, Sigma-Aldrich, Inc. Ontario, Canada) and iron oxide (iron (III) oxide, nano-powder, <50 nm particle size, Sigma-Aldrich, Inc.). For in situ real-time measurements of ambient aerosols, an electrically conductive tube ~122 cm in length was used to direct the ambient air through the gas flow cuvette placed on the stage of the DIHM instrument. The outflow (1.7 L/min) from the cuvette passed to the SMPS and the OPS. The coupling of the SMPS and the OPS with the DIHM allowed determining the aerosol size distribution of the particles imaged by DIHM in real time.

The original colloid solution of 100 nm PSL spheres (2% wt) was provided with a calibrated diameter of 101 ± 7 nm and the size of 200 nm PSL (10% wt) standard particles with a calibrated diameter of 188 ± 4 nm, supplied by Sigma-Aldrich, respectively. The original colloid solution was diluted for aerosolization and imaging in the moving airflow stream by mixing 100 μL of the original colloid solution of the PSL spheres with 10 mL of Milli-Q water. The resultant solution was aerosolized using the aerosol generation unit (solution flow rate of 0.25 ml/minute) and passed through the gas flow cuvette installed in the DIHM instrument with a final flow rate of 1.7 L/min.

We investigated the size and shape of the PSL spheres in the colloid solution of PSL standard particles and 1 : 100 mixtures with Milli-Q water at room temperature. In addition, the mixture of iron oxide nanoparticles and 200 nm standard PSL particles were analysed by directly using DIHM. A 20 μL aqueous solution of PSL particles (100 and 200 nm) and mixed samples were deposited on the microscope slide to record the direct holograms by DIHM, respectively. Next, standard-size synthetic powder materials (zinc oxide and iron oxide) were imaged by DIHM directly on a microscope slide. To validate the size, shape, and morphology of the PSL, mixed samples and particles in the snow were visualized in the holograms and compared with the morphology visualized using S/TEM. Half of the liquid samples were analysed by S/TEM at the Facility of electron microscopy at McGill University and the remaining half samples were studied using Nano-DIHM. The S/TEM imaging techniques are discussed in our previous papers[15,16]. The additional details of each experiment, such as sample information, flow rates, and the recording parameters of DIHM, are given in Table 1.

We evaluated the changing refractive index measurement of the suspended solution. Glycerin drops were prepared by adding 100 μl of glycerin (refractive index, 1.4729, Sigma-Aldrich, Inc.) to 1 ml of type F microscope oil (refractive index, 1.518 at 23 °C, Olympus, immoil-f30cc, Tokyo, Japan). This mixture was then vortexed for 15 min and a small sample of the resulting suspension was placed on a microscope slide. A previous study suggests that this method produced glycerin drops in the range of 1–10 μm in diameter[78].

## Data availability
All the numerical data generated in this study are available within the paper in graphical representation. The data that support the findings of this study are available from the corresponding author upon reasonable request.

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

## Acknowledgements

This work was supported by the Natural Sciences and Engineering Research Council of Canada (NSERC), National Research Council (NRC), NSERC CREATE Mine of Knowledge, NSERC CREATE Pure, Pôle de recherche et d'innovation en matériaux avancés du Québec (PRIMA Quebec), and Environment and Climate Change Canada (ECCC). We are thankful to Dr. Jim Avik Ghoshdastidar, University of Toronto, and Dr. Amit Kumar Pandit, National Institute of Aerospace, Hampton Virginia, USA, for his critical review of the manuscript and Mr. Ryan Hall for proofreading our manuscript. We also thank Dr. Justin Jacquot and Dr. Deokhyeon Kwon for their help in installing the Holography setup in the laboratory. Finally, we are very grateful to Professor Hans Juergen Kreuzer, Dalhousie University, for his guidance in hologram reconstruction and verification of our results.

## Author contributions

D.P. performed the experimental work and data analysis and wrote the first draft. D.P., Y.N., T.C.P. and P.A.A. wrote and revised the manuscript. P.A.A. supervised D.P. and Y.N. P.A.A. wrote the original proposal, which was the basis of this original project and supported the project financially.

## Competing interests

The authors declare no competing interests.
