## [Peer Review File · Communications Chemistry]

Reviewers' comments:

Reviewer #1 (Remarks to the Author):

Title: Advancing the Science of Dynamic Airborne Nanosize Particles using Optimized Digital Holographic Microscopy

General comment:

This manuscript demonstrates the acquisition of physical properties of airborne nanoparticles by using a digital in-line holographic microscopy (DIHM). 3D distribution, phase, morphology, and refractive index of nanoparticles were analyzed quantitatively with high-resolution. However, the following points need to be addressed to make this manuscript more structured and informative.

Detailed comments:

1. Page 23, Line 401-402: The authors mentioned that the lateral and depth resolutions of the proposed experimental setup was 271 nm and 362 nm, respectively. The resolutions were evaluated by theoretical equations 4-6. However, the actual resolutions of holographic images recorded for airborne nanoparticles were not tested in this study. The authors should verify the lateral and depth resolutions of the proposed DIHM technique.
2. Figure 2h shows 3D orientation of PSL spheres reconstructed from a single hologram. However, the numerical reconstruction procedure may cause measurement errors in the 3D position and shape of the particles. The measurement accuracy of DIHM may depend on the propagation distance (z) from the focal plane. The authors should estimate the errors between the actual and reconstructed depth positions of particles.
3. Page 17, Line 289-291: The authors mentioned that 3D reconstruction with high-resolution was validated in Fig. 2 by visualizing the reconstructed image planes at six different depth positions. However, Fig. 2 is just a reconstructed result of a single hologram rather than a validation. Thus, the authors should carry out additional validation as mentioned in previous comments, or delete the sentence.
4. Page 5, Line 84-85: The authors mentioned, "Here we developed a Nano-DIHM technique capable of characterizing numerous nanosized particles in a volume of air without single-particle trapping". Conventional DIHM setup with an objective lens has limitations in the field-of-view and depth-of-field. Thus, it is challenging to capture holograms of airborne particles moving fast along the out-of-plane direction in an image plane. To solve this problem, the authors used a cuvette to make aerosol flow. However, it looks like that a gas flow cuvette was also utilized as a sample trapper to place ambient aerosols at a desirable distance from a pinhole. The authors should explain the conceptual difference between conventional single-particle trapping and a gas flow cuvette in more detail.
5. Page 16, Line 265-267: The authors mentioned, "Nano-DIHM efficiently resolves the size and shape of PSL particles in moving air, demonstrating the newly developed technique's ability to describe aerosol dynamics with rapid sequential hologram acquisitions". However, the dynamic motions of airborne nanoparticles might be challenging to analyze due to the limited FOV of the proposed experimental setup. In addition, there is no experiment result about trajectories of aerosols, such as particle tracing. The authors need to provide additional information to support that the proposed method can analyze aerosol dynamics.

Reviewer #2 (Remarks to the Author):

The work proposed by the authors for a publication is interesting and can be useful to the scientific community. This is a smart approach to determine the geometric and physical parameters of micro and nanoparticles measured in situ as well as a determination of their size distribution function. For that, the authors used available instrumentation and software to carry out their study on different types of particles in solid, liquid or gaseous medium.

The reviewer recognizes the quantity and quality of the experimental work, however, the writing of the article is made very difficult by its structure and the sequencing of the paragraphs which really lack a common thread. For example, a clear and concise objective is not really presented and the presentation of the results before having explained the underlying theory (Digital Inline Holographic Microscopy), the instrumentation and the parameters extraction algorithms seems to me counterproductive and seriously handicaps the interest of the work.

In addition, the desire to insert the interest of this study in the problem of SARS COV-2 seems risky to me since this virus has a size around 60-100 nm and cannot be studied or visualized with the used instrumentation and its specific wavelength.

I recommend a complete rewrite of the article to make it more in line with the editorial rules in scientific journals.

Reviewer #3 (Remarks to the Author):

The main claim of this paper is that they have developed a nano-DIHM that can image nano-size particles suspended in the air or liquids. This claim, if it is true, would be a major breakthrough in DIHM technology. However, after having gone through this entire, I found such a claim is not substantiated. Here are specific comments supporting my assessment

1. The paper does not provide any technical details to show how they can break the barrier of diffraction limit to see the nano-size particles using DIHM. It appears to me that the entire system is still diffraction-limited as they mentioned in their paper that DIHM resolution is around ~300 nm. Numerous DIHM papers have shown imaging particles in this range. Claiming their sensor is a nano-particle sensor is an overstatement.

2. I don't see a clear demonstration of the ability to image nanoparticles from any of the figures. Most of the figures are showing the particles of around micron size or above. Only figures 1-2 seem to be showing some submicron particles. But the image quality is very poor. I don't see a clear hologram signature associated with each nanoparticle. The small dots shown in these images look like single-pixel noise. There is no detailed comparison between holographic images and SEM images of nanoparticles even if the authors claim they use nanoparticles of 50 nm. I suggest the authors check the following paper (the only paper I know of that clearly demonstrated nanoparticle image capability to get a sense of what level of quality images they need to provide to show nanoparticle imaging capability.

Mudanyali, O., McLeod, E., Luo, W., Greenbaum, A., Coskun, A. F., Hennequin, Y., ... & Ozcan, A.

(2013). Wide-field optical detection of nanoparticles using on-chip microscopy and self-assembled nanolenses. *Nature Photonics*, 7(3), 247.

3. The paper failed to review any past effort in the field of DIHM on the development of nanoparticle sensing ability. The authors show little knowledge of the development of DIHM in the past.

Prof. Parisa Angeline Ariya

Department of Chemistry and Department of Atmospheric and Oceanic Sciences

McGill University, 801 Sherbrooke St. West Montreal, PQ, Canada

E-mail: parisa.ariya@mcgill.ca

RE: Re-submission of the manuscript entitled "Advancing the Science of Dynamic Airborne Nanosized Particles using Optimized Digital Holographic Microscopy"

July 09, 2021

To: Editor/reviewers,
Communications Chemistry: Nature

Dear editor/colleagues,

Please find an attached manuscript entitled "Advancing the Science of Dynamic Airborne Nanosize Particles using Optimized Digital Holographic Microscopy" by Devendra Pal, Yevgen Nazarenko, Thomas C. Preston, and Parisa A. Ariya, for consideration for publication in your journal.

This manuscript was submitted last year. Initially, the manuscript was rejected. We received mixed comments, yet all had raised some good suggestions and some constructive critics and reservations.

We would like to sincerely thank all reviewers for their time, suggestions and consideration. We very much appreciate critical comments and suggestions. We have considered them all and have made the requested significant modifications to the manuscript accordingly. We have also performed additional experiments improving the quality of data on two fronts: 1) targeted experiments to provide the evidence of nanosized particles, and 2) improvement of image reconstructions and temporal resolutions, providing solid evidence for Nano-DIHM being an in-situ and real-time aerosols size, shape and phase observation of all sizes including *airborne nanoparticles*.

We have also performed the following additional experiments:

- **Demonstrating Nano-DIHM in-situ and real-time particle observation in the air:** Tracking airborne PSL (100 nm and 200 nm) particle. Note that a suite of complementary experiments has also shown its effectiveness for nanosize particle detection in water and snow.
- **Intercomparison of Nano-DIHM data with S/TEM using similar samples:** Validation of the nano-DIHM particle size and shape analysis with S/TEM technique. It confirms that Nano-DIHM can be operated in dynamic real-time and in situ, and in stationary manners.

- **Mixed size particle experiments:** Validation of Nano-DIHM analysis with S/TEM.
- **Automation:** In this paper, we also included advances in software automation (deep-learning). It now enables a 3rd digit accuracy in the numerical reconstruction software that provides high-resolution images that can be done remotely. In addition, we are now working to fully automatize the Stingray software¹, and significant progress has been made (automation details provided in supplementary information Text S4). An example of the detection of PSL particle aerosol is given in the Supplementary Information section (Figure S10).

We have thus included a response to all suggestions, additional experiments and the revised manuscript for your reconsideration. This paper is drastically superior in scientific quality, providing the evidence for Nano-DIHM being an option for dynamic aerosols observation of all sizes, shapes, and phases, including airborne nanoparticles in dynamic media.

Thanks very much in advance for your time and reconsideration. If you require any further information, please feel free to contact me.

Cordially,

Parisa A. Ariya

James McGill Chair of Chemistry and Atmospheric and Oceanic Sciences

McGill

Department of
**Atmospheric and
Oceanic Sciences**

Département des
**sciences atmosphériques
et océanographiques**

Reply

For your convenience, the reviewers' comments are italicized, and our responses are in regular font.

Reviewer #1

General comment:

"This manuscript demonstrates the acquisition of physical properties of airborne nanoparticles by using a digital in-line holographic microscopy (DIHM). 3D distribution, phase, morphology, and refractive index of nanoparticles were analyzed quantitatively with high-resolution. However, the following points need to be addressed to make this manuscript more structured and informative."

Response: Thank you very much for your comments. As it is shown in the following sections, we have considered all your suggestions, and we trust that we have addressed them accordingly.

Detailed comments:

1. "Page 23, Line 401-402: The authors mentioned that the lateral and depth resolutions of the proposed experimental setup was 271 nm and 362 nm, respectively. The resolutions were evaluated by theoretical equations 4-6. However, the actual resolutions of holographic images recorded for airborne nanoparticles were not tested in this study. The authors should verify the lateral and depth resolutions of the proposed DIHM technique."

Response: Thank you for your comment. Your point is well-taken. In the revised manuscript, we significantly improved the quality of data on two fronts:

- 1) A targeted experiment to provide the evidence of nanosized particles and
- 2) Improvement of image reconstructions and temporal resolutions. In addition, the structure of the paper has been changed, so are the Figures and Tables that are replaced, and others are relocated.
 - Figure 1(i and k) is an example of the real-time observation of lateral resolution of Nano-DIHM is ~ 200 nm in the revised manuscript. In addition, the recovery of shape, size, and phase of various PSL spheres (100 nm and 200 nm) in both dynamic (airborne phase) and stationary (aqueous mode) medium confirmed the ability of Nano-DIHM to detect the particles below lateral resolution.
 - The limited resources and lack of optical targets with 200 nm lateral spacing constrained us to test the standard. However, STEM images of duplicate samples of particles confirmed the observation of nanosized particles made by Nano-DIHM. All the new results are discussed in the revised manuscript in the following order:
 - 1) Observation of airborne PSL (100 nm and 200 nm) particles and their dynamic trajectories;

- 2) Detection of the 100 nm and 200 nm PSL (colloid solution) deposited on the microscopy slide;
- 3) Distinction of the PSL from iron oxide nanoparticles in the mixed samples, and
- 4) An example of real-time measurement of particles in ambient air, as well as particles in the snow (Supplementary Information (SI), Text S1) in real time and in situ.
- The demonstration of the accuracy of measurement of refractive indices presented in the supplementary information (SI, Text S2).

Please note that the improvement mentioned above in the quality of data of nanosized particles was achieved by:

1) Improved experiment approach:

- The hologram has been recorded at the tipping point of the pinhole (source). The pinhole and the flow tube (cuvette) or microscopic slide were nearly touching each other. In DIHM, the shorter the source-to-sample distance, the higher the magnification, and hence a higher resolution can be achieved. This experimental recording process of holograms enabled us to overcome the diffraction barrier, allowing measurement of the nanosized particles ≤ 200 nm. Note that several studies using near-field optical microscopy have successfully captured the nanoscale objects, overcoming the diffraction limit of the optical system²⁻⁸.
- Indeed, as the distance between the source and the sample gets smaller, the object vibrations and noise are getting more extensive due to the higher magnification. It blurs out the fine interference fringes and reduces the potentially achievable high resolution. Indeed, it is a challenge. However, experimentally, to overcome this significant challenge and compensate for the vibrations and noise, a short-time acquisition sequence of the hologram was used and recorded^{9,10}. Later, numerical reconstruction methods with deblurring techniques have been demonstrated that in the motion-deblurring photo-image analysis permits achieving a higher resolution by a factor of 2 or more^{9,11,12}. Hence, the major challenge has been shown to be overcome.

Hence, by building on the past research, this major challenge has been shown to be overcome, as seen in the manuscript.

2) Numerical reconstruction process:

In this study, the Octopus/Stingray software^{1,12} was used to reconstruct the recorded holograms. The software is based on a basic patented algorithm¹³, which can effortlessly achieve the lateral resolution on the order of half-wavelength ($\lambda/2$) of source and depth resolution on the order of one wavelength (λ)¹⁴. Furthermore, as shown in detail elsewhere,^{9,11,14,15} a higher resolution was achieved using multiple deconvolutions during the reconstruction: one for the illumination system (pinhole) and the second one for the finite numerical aperture of the recording screen

(CCD/CMOS). In previous digital holography research, pixel super-resolution by source shifting numerical reconstruction methods has been used to achieve high-resolution images¹⁶⁻¹⁸.

- 3) **Experiments in two media:** In addition to air, several holograms are formed in an aqueous medium (water), and thus the wavelength of the laser is reduced^{19,20} from $\lambda = 405$ nm to ~ 300 nm.

In the revised manuscript, the above description is included in the method section from Line 594-521.

"2. Figure 2h shows 3D orientation of PSL spheres reconstructed from a single hologram. However, the numerical reconstruction procedure may cause measurement errors in the 3D position and shape of the particles. The measurement accuracy of DIHM may depend on the propagation distance (z) from the focal plane. The authors should estimate the errors between the actual and reconstructed depth positions of particles."

Response: Thank you very much for your comment. Indeed, the three-dimensional distribution of the particles are reconstructed from a single 2D hologram suffers from a limited depth of focus, and the reconstructed particles do not appear well localized and are prolonged in the z -direction⁹. The error between the actual and reconstruction depth of the field were observed within ± 5 %.

"3. Page 17, Line 289-291: The authors mentioned that 3D reconstruction with high resolution was validated in Fig. 2 by visualizing the reconstructed image planes at six different depth positions. However, Fig. 2 is just a reconstructed result of a single hologram rather than a validation. Thus, the authors should carry out additional validation as mentioned in previous comments, or delete the sentence."

Response: Thank you for your comment. Fig. 2 (a-f) in the previous manuscript illustrated the particle distribution of a single hologram at different reconstructions at different positions (Z). It demonstrates that the distribution of particles changes with changing the reconstruction position. However, during the 3D reconstruction process from a single 2D hologram, most of the particles have been recovered, given the boundary condition. Therefore, we replaced Fig. 2(a-f) from the previous manuscript by 3D distributions of 100 nm and 200 nm PSL airborne particle size distribution in Figure 2(a-d) and Figure S3(a-f) in the revised manuscript.

"We have performed several 3D reconstructions of different holograms, shown in Figure 2 (a-c), Figure S2 and Figure S3. The median and mean particle dimensions varied within ± 5 % from one hologram to the next, within the reconstruction process of several holograms. This uncertainty likely arises during hologram reconstruction due to the input digital image data adding noises during the recording process.

"4. Page 5, Line 84-85: The authors mentioned, "Here we developed a Nano-DIHM technique capable of characterizing numerous nanosized particles in a volume of air without single-particle trapping". Conventional DIHM setup with an objective lens has limitations in the field-of-view and depth-of-field. Thus, it is challenging to capture holograms of airborne particles moving fast along the out-of-plane direction in an image plane. To solve this problem, the authors used a cuvette to make aerosol flow. However, it looks like that a gas flow cuvette was also utilized as a sample trapper to place ambient aerosols at a desirable distance from a pinhole. The authors should explain the conceptual difference between conventional single-particle trapping and a gas flow cuvette in more detail."

Response: Thank you very much for your comment. In the revised manuscript, we structured the introduction according to your suggestions:

- 1) We highlighted the differences between conventional methods of optical microscopy with digital in-line holography.
- 2) We provided evidence of the advantage of coupling a cuvette with Nano-DIHM for direct aerosol measurements vs. sample trapping methods, like particle deposition or optical trapping of aerosol.

In the revised manuscript, a detailed description is given in Line 56-93:

"Significant advances in microscopy during the recent decades have enabled researchers to observe individual nanoparticles^{2,3}, from near-field optical microscopy⁶⁻⁸, super-resolution microscopy^{4,21,22}, atomic force microscopy²³, electron microscopy²⁴ and more recent imaging techniques^{5,16,25-27}. Among them, S/TEM enables acquiring accurate information on particle phase and morphology in stationary mode²⁸. Bright-field and dark-field optical microscopy provide two-dimensional information on particles, albeit in a limited depth interval within samples^{5,29}. A challenge with conventional light microscopy methods is that these methods work in fixed imaging planes²⁹, which precludes determining aerosol dynamics, phase, and 3D morphology of aerosols²⁹. However, the 3D structure information can be obtained using Fourier ptychography³⁰, optical diffraction tomography³¹ or by scanning the whole sample/particle volume using confocal imaging³². All these existing microscopy techniques have significant advantages, yet they cannot track moving particles in-situ or in real time, precluding their application to dynamic media, such as air.

Here, we provide an alternative approach of Nano-DIHM. The Nano-DIHM consists of a holographic microscope and a gas flow tube that allows airborne particles to pass through the imaging volume of the DIHM and exit or circulate inside the volume, allowing observation of single particles or ensembles of particles in real time. Nano-DIHM directly acquires data on interference patterns of the incident and scattered light with a light-sensitive matrix, without any lenses or objective²⁰. The recorded interference pattern referred to as a hologram is numerically reconstructed using an Octopus/Stingray software based on a patented algorithm¹³ to recover the object information^{12,20,33}. To date, digital holography setups could merely characterize particles held in electrodynamic^{34,35} or optical traps^{36,37} or particles deposited on a substrate^{16,17,20,33,38,39}. Note that optical trapping of airborne particles requires optical tweezers, trapping only a single particle confined to the field of view^{37,40,41}. The Nano-DIHM has a larger field of view up to several square centimetres ($\sim 1.27 \text{ cm}^2$, 2048*2048 pixel, 5.5 μm each

pixel size). In this study, the Nano-DIHM field of view up to $\sim 40 \text{ mm}^2$ enables observation of moving objects, in contrast to conventional optical microscopy that uses lenses and has smaller fields of view (a few microns)^{5,16}. Until now, investigations of airborne particles by DIHM have been confined to relatively large particles ($> 1 \mu\text{m}$)⁴²⁻⁴⁷. We show that Nano-DIHM can detect nanosized objects in 2D and 3D space for dynamic (air), aqueous (water) and solid (powder) media. We were able to track individual airborne nanoparticles directly and quantify each particle's dimensions in situ and in real time. The Nano-DIHM enabled us to record 6D spatial motion of aerosol particles (3D for the position of each particle in 3D space) and the dimensions of each particle as it is orientated in 3D space^{38,48}. A critical advantage of Nano-DIHM is that during reconstruction, a single 2D hologram can produce a 3D image of the objects without any loss of resolution^{20,38}. The Octopus/Stingray software allows real-time or offline reconstruction with a temporal resolution on the order of milliseconds (62.5 ms), and it can be improved to microsecond-scale temporal resolution, depending on the framerate of the camera^{37,44,49,50}.”

"5. Page 16, Line 265-267: The authors mentioned, "Nano-DIHM efficiently resolves the size and shape of PSL particles in moving air, demonstrating the newly developed technique's ability to describe aerosol dynamics with rapid sequential hologram acquisitions". However, the dynamic motions of airborne nanoparticles might be challenging to analyze due to the limited FOV of the proposed experimental setup. In addition, there is no experiment result about trajectories of aerosols, such as particle tracing. The authors need to provide additional information to support that the proposed method can analyze aerosol dynamics."

Response: Thank you for the comment. In the revised manuscript, we provided the particle's trajectory motion within 62.5-millisecond intervals for several hologram acquisitions in SI-Giff 1 and SI-Giff 2. The results are stated in the revised manuscript from Line 242-253 as:

“Figure 2 (a-b) and Figure S3 (a-f) illustrate the 6D data for PSL spheres, including the orientation of particles in 3D space (Figure 2 (a) and Figure S3 (a)) and the three dimensions of each particle (width, height, and length) in a single hologram (Figure 2 (b) and Figure S3 (b)). The dynamic 4D trajectories (3D positions and 1D time) of 100 nm PSL spheres are provided in the SI-Giff 1 and SI-Giff 2 with a temporal resolution of 62.5 ms. The SI-Giff 1 is a zoomed-in image of SI-Giff 2. All the reconstructions of the holograms were performed at the same reconstruction distance (z) and then processed to create the Giffs. As depicted in SI-Giff 1, the dark red particles are in the focus of the reconstruction plane, while some of the particles (green/blue) are a little bit out of focus at the exact reconstruction distances due to the finite depth of field of the objective. The out-of-focus particles can be further modified by changing the reconstruction position. As such, we demonstrated the ability of the Nano-DIHM to visualize particle trajectories.”

Reviewer #2:

"The work proposed by the authors for a publication is interesting and can be useful to the scientific community. This is a smart approach to determine the geometric and physical parameters of micro and nanoparticles measured in situ as well as a determination of their size distribution function. For that, the authors used available instrumentation and software to carry out their study on different types of particles in solid, liquid or gaseous medium.

The reviewer recognizes the quantity and quality of the experimental work, however, the writing of the article is made very difficult by its structure and the sequencing of the paragraphs which really lack a common thread. For example, a clear and concise objective is not really presented and the presentation of the results before having explained the underlying theory (Digital Inline Holographic Microscopy), the instrumentation and the parameters extraction algorithms seems to me counterproductive and seriously handicaps the interest of the work.

In addition, the desire to insert the interest of this study in the problem of SARS COV-2 seems risky to me since this virus has a size around 60-100 nm and cannot be studied or visualized with the used instrumentation and its specific wavelength.

I recommend a complete rewrite of the article to make it more in line with the editorial rules in scientific journals."

Response: Thank you very much for your comments on the quality of data, quality of experiments and the significant contribution of the article in scientific research. As you can see in the following section, we have considered all your suggestions, carefully. We trust that we have addressed them, accordingly.

We have also performed following additional experiments:

- **Demonstrating nano-DIHM in-situ and real-time observation:** Tracking airborne PSL (100 nm and 200 nm) particle (Line 117-241).
- **Intercomparison of Nano-DIHM data with S/TEM using similar samples:** validation of the nano-DIHM particle size analysis (Line 254-296)
- **Mixed size particle experiments:** Validation of Nano-DIHM analysis with HR-S/TEM (Line 307-324)
- **Automation:** Enabling a 3rd digit accuracy in the numerical reconstruction software that provides high-resolution images. We are also working to fully automatized the Stingray software¹, and significant progress has been made. An example of detection PSL particle aerosol is given in the Supplementary Information section (Figure S10). In addition, the automation process of stingray software is discussed in supplementary text 4.

- **Mention of bioaerosols:** We understand your comment and thus decided to delete all referrals to bioaerosols in this paper. However, the reason we mentioned the bioaerosols

was the fact that we had measured standard aerosolized MS2 bacteriophage, an airborne virus, has been detected by nano-DIHM in the laboratory. Here is an example. In short, we removed all mentions of aerosols.

Figure 1 Holographic reconstruction of MS2 bacteriophage and validation made by S/TEM. (a, b) Phase images of MS2 bacteriophage and (c, d) Results made by S/TEM.

Reviewer #3:

"The main claim of this paper is that they have developed a nano-DIHM that can image nanosize particles suspended in the air or liquids. This claim, if it is true, would be a major breakthrough in DIHM technology. However, after having gone through this entire, I found such a claim is not substantiated. Here are specific comments supporting my assessment."

Response: Thank you very much for your comments on the quality of data and the significant contribution of the article in DIHM technology. We have performed more experiments and extensive reconstruction of holograms. As you can see in the following section, we have considered all your suggestions, and we trust that we have addressed them accordingly.

"1. The paper does not provide any technical details to show how they can break the barrier of diffraction limit to see the nanosize particles using DIHM. It appears to me that the entire system is still diffraction-limited as they mentioned in their paper that DIHM resolution is around ~300 nm. Numerous DIHM papers have shown imaging particles in this range. Claiming their sensor is a nanoparticle sensor is an overstatement."

Response: Thank you for your comment. Indeed, the calculated lateral resolution in this study was 271 nm. Three processes have been followed to achieve the better lateral resolution than the diffraction limits of the system:

1) Improved experiment approach:

- The hologram has been recorded at the tipping point of the pinhole (source). The pinhole and the flow tube (cuvette) or microscopic slide were nearly touching each other. In DIHM, the shorter the source-to-sample distance, the higher the magnification, and hence a higher resolution can be achieved. This experimental recording process of holograms enabled us to overcome the diffraction barrier, allowing measurement of the nanosized particles ≤ 200 nm. Note that several studies using near-field optical microscopy have successfully captured the nanoscale objects, overcoming the diffraction limit of the optical system²⁻⁸.
- Indeed, as the distance between the source and the sample gets smaller, the object vibrations and noise are getting more extensive due to the higher magnification. It blurs out the fine interference fringes and reduces the potentially achievable high resolution. Indeed, it is a challenge. However, experimentally, to overcome this significant challenge and compensate for the vibrations and noise, a short-time acquisition sequence of the hologram was used and recorded^{9,10}. Later, numerical reconstruction methods with deblurring techniques have been demonstrated that in the motion-deblurring photo-image analysis permits achieving a higher resolution by a factor of 2 or more^{9,11,12}. Hence, the major challenge has been shown to be overcome.

Hence, by building on the past research, this major challenge has been shown to be overcome, as seen in the manuscript.

2) Numerical reconstruction process:

In this study, the Octopus/Stingray software^{1,12} was used to reconstruct the recorded holograms. The software is based on a basic patented algorithm¹³, which can effortlessly achieve the lateral resolution on the order of half-wavelength ($\lambda/2$) of source and depth resolution on the order of one wavelength (λ)¹⁴. Furthermore, as shown in detail elsewhere,^{9,11,14,15} a higher resolution was achieved using multiple deconvolutions during the reconstruction: one for the illumination system (pinhole) and the second one for the finite numerical aperture of the recording screen (CCD/CMOS). In previous digital holography research, pixel super-resolution by source shifting numerical reconstruction methods has been used to achieve high-resolution images¹⁶⁻¹⁸.

3) Experiments in two media:

In addition to air, several holograms are formed in an aqueous medium (water), and thus the wavelength of the laser is reduced^{19,20} from $\lambda = 405$ nm to ~ 300 nm.

In the revised manuscript, the above description is included in the method section from Line 494-521.

"2. I don't see a clear demonstration of the ability to image nanoparticles from any of the figures. Most of the figures are showing the particles of around micron size or above. Only figures 1-2 seem to be showing some submicron particles. But the image quality is very poor. I don't see a clear hologram signature associated with each nanoparticle. The small dots shown in these images look like single-pixel noise. There is no detailed comparison between holographic images and SEM images of nanoparticles even if the authors claim they use nanoparticles of 50 nm. I suggest the authors check the following paper (the only paper I know of that clearly demonstrated nanoparticle image capability to get a sense of what level of quality images they need to provide to show nanoparticle imaging capability.)"

Mudanyali, O., McLeod, E., Luo, W., Greenbaum, A., Coskun, A. F., Hennequin, Y., ... & Ozcan, A. (2013). Wide-field optical detection of nanoparticles using on-chip microscopy and self-assembled nanolenses. Nature Photonics, 7(3), 247.

Response: Thank you for the comment and your suggestions. In the revised manuscript, we added the proposed references, and we have also provided the additional experiments by Nano-DIHM and comparison with S/TEM.

We have also performed following the additional experiments:

- **Demonstrating nano-DIHM in-situ and real-time observation:** Tracking airborne PSL (100 nm and 200 nm) particle (Line 117-241).
- **Intercomparison of Nano-DIHM data with S/TEM using similar samples:** Validation of the nano-DIHM particle size analysis (Line 254-269)

- **Mixed size particle experiments:** Validation of Nano-DIHM analysis with S/TEM Line 307-324)

Please note that in addition to these experiments, we have achieved:

- **Automation:** Enabling a 3rd digit accuracy in the numerical reconstruction software that provides high-resolution images. We are also working to fully automatized the Stingray software¹, and significant progress has been made. An example of detection PSL particle aerosol is given in the Supplementary Information section (Figure S10). In addition, the automation process of stingray software is discussed in supplementary text 4.

The outcome of the above-listed experiment suggested that the Nano-DIHM successfully recovered the size, shape, phase, and morphology of nanosized objects in dynamic (airborne), aqueous (water), and solid (powder) media.

“3. The paper failed to review any past effort in the field of DIHM on the development of nanoparticle sensing ability. The authors show little knowledge of the development of DIHM in the past.”

Response: Thank you very much for your comment. In the revised manuscript, we have provided that the past literature (Most recent references: ^{16,25-27,34,35,16,17,20,33,36-39,42-47}) has been made for DIHM development. However, we only focused on the appropriate literature of DIHM application for airborne particle detection. Therefore, we restructured the introduction and highlighted the DIHM application and the past effort for DIHM application for airborne particle detection and approaches to overcome the diffraction limit.

In the revised manuscript, a detailed description is given in Line 56-93:

References:

- 1 imaging, D. i. Stingray Software User Guide. (2018).
- 2 Won, R. Eyes on super-resolution. *Nature Photonics* **3**, 368-369, doi:10.1038/nphoton.2009.103 (2009).
- 3 Zhuang, X. Nano-imaging with STORM. *Nature Photonics* **3**, 365-367, doi:10.1038/nphoton.2009.101 (2009).
- 4 Huang, B., Babcock, H. & Zhuang, X. Breaking the diffraction barrier: super-resolution imaging of cells. *Cell* **143**, 1047-1058, doi:10.1016/j.cell.2010.12.002 (2010).
- 5 Liu, W. *et al.* Breaking the Axial Diffraction Limit: A Guide to Axial Super-Resolution Fluorescence Microscopy. *Laser & Photonics Reviews* **12**, 1700333, doi:<https://doi.org/10.1002/lpor.201700333> (2018).
- 6 Ozcan, A. *et al.* Differential near-field scanning optical microscopy. *Nano letters* **6**, 2609-2616 (2006).
- 7 de Lange, F. *et al.* Cell biology beyond the diffraction limit: near-field scanning optical microscopy. *J Cell Sci* **114**, 4153-4160 (2001).
- 8 Betzig, E. & Chichester, R. J. Single Molecules Observed by Near-Field Scanning Optical Microscopy. *Science* **262**, 1422, doi:10.1126/science.262.5138.1422 (1993).
- 9 Latychevskaia, T., Gehri, F. & Fink, H.-W. Depth-resolved holographic reconstructions by three-dimensional deconvolution. *Optics Express* **18**, 22527-22544, doi:10.1364/OE.18.022527 (2010).

- 10 Germann, M., Latychevskaia, T., Escher, C. & Fink, H.-W. Pulsed electron holography. *Applied Physics Letters* **102**, 203115 (2013).
- 11 Latychevskaia, T. & Fink, H.-W. Resolution enhancement in in-line holography by numerical compensation of vibrations. *Optics express* **25**, 20109-20124 (2017).
- 12 imaging, D. i. Octopus Software User Guide. (2018).
- 13 Kreuzer, H. J. US. Patent 6411406 B1, Canadian Patent CA 2376395. (2002).
- 14 Nickerson, B. S. & Kreuzer, H. J. Deconvolution for digital in-line holographic microscopy. *Advanced Optical Technologies* **2**, 337-344, doi:doi:10.1515/aot-2013-0030 (2013).
- 15 Kanka, M., Riesenberger, R. & Kreuzer, H. J. Reconstruction of high-resolution holographic microscopic images. *Optics Letters* **34**, 1162-1164, doi:10.1364/OL.34.001162 (2009).
- 16 Mudanyali, O. *et al.* Wide-field optical detection of nanoparticles using on-chip microscopy and self-assembled nanolenses. *Nature Photonics* **7**, 247-254, doi:10.1038/nphoton.2012.337 (2013).
- 17 Mudanyali, O., Bishara, W. & Ozcan, A. Lensfree super-resolution holographic microscopy using wetting films on a chip. *Optics express* **19**, 17378-17389 (2011).
- 18 Bishara, W. *et al.* Holographic pixel super-resolution in portable lensless on-chip microscopy using a fiber-optic array. *Lab on a Chip* **11**, 1276-1279 (2011).
- 19 Jericho, M. H., Kreuzer, H. J., Kanka, M. & Riesenberger, R. Quantitative phase and refractive index measurements with point-source digital in-line holographic microscopy. *Applied Optics* **51**, 1503-1515, doi:10.1364/AO.51.001503 (2012).
- 20 Garcia-Sucerquia, J. *et al.* Digital in-line holographic microscopy. *Applied optics* **45**, 836-850 (2006).
- 21 Huang, F. M. & Zheludev, N. I. Super-resolution without evanescent waves. *Nano letters* **9**, 1249-1254 (2009).
- 22 Kner, P., Chhun, B. B., Griffis, E. R., Winoto, L. & Gustafsson, M. G. Super-resolution video microscopy of live cells by structured illumination. *Nature methods* **6**, 339-342 (2009).
- 23 Ebenstein, Y., Nahum, E. & Banin, U. Tapping mode atomic force microscopy for nanoparticle sizing: tip-sample interaction effects. *Nano Letters* **2**, 945-950 (2002).
- 24 Feinstone, S. M., Kapikian, A. Z. & Purcell, R. H. Hepatitis A: detection by immune electron microscopy of a viruslike antigen associated with acute illness. *Science* **182**, 1026-1028 (1973).
- 25 Hong, X. *et al.* Background-free detection of single 5 nm nanoparticles through interferometric cross-polarization microscopy. *Nano letters* **11**, 541-547 (2011).
- 26 Ray, A. *et al.* Holographic detection of nanoparticles using acoustically actuated nanolenses. *Nature Communications* **11**, 171, doi:10.1038/s41467-019-13802-1 (2020).
- 27 Warnasooriya, N. *et al.* Imaging gold nanoparticles in living cell environments using heterodyne digital holographic microscopy. *Optics express* **18**, 3264-3273 (2010).
- 28 Rangel-Alvarado, R. B., Nazarenko, Y. & Ariya, P. A. Snow-borne nanosized particles: Abundance, distribution, composition, and significance in ice nucleation processes. *Journal of Geophysical Research: Atmospheres* **120**, 11,760-711,774 (2015).
- 29 Murphy, D. B. *Fundamentals of light microscopy and electronic imaging*. (John Wiley & Sons, 2002).
- 30 Tian, L. *et al.* Computational illumination for high-speed in vitro Fourier ptychographic microscopy. *Optica* **2**, 904-911 (2015).
- 31 Horstmeyer, R., Chung, J., Ou, X., Zheng, G. & Yang, C. Diffraction tomography with Fourier ptychography. *Optica* **3**, 827-835 (2016).
- 32 Jeong, H.-j., Yoo, H. & Gweon, D. High-speed 3-D measurement with a large field of view based on direct-view confocal microscope with an electrically tunable lens. *Optics express* **24**, 3806-3816 (2016).
- 33 Berg, M. J. & Holler, S. Simultaneous holographic imaging and light-scattering pattern measurement of individual microparticles. *Optics letters* **41**, 3363-3366 (2016).

- 34 Barnes, M. D., Lermer, N., Whitten, W. B. & Ramsey, J. M. A CCD based approach to high-precision size and refractive index determination of levitated microdroplets using Fraunhofer diffraction. *Review of Scientific Instruments* **68**, 2287-2291, doi:10.1063/1.1148134 (1997).
- 35 Steimer, S. S. *et al.* Electrodynamic balance measurements of thermodynamic, kinetic, and optical aerosol properties inaccessible to bulk methods. *Atmos. Meas. Tech.* **8**, 2397-2408, doi:10.5194/amt-8-2397-2015 (2015).
- 36 Carruthers, A. E., Walker, J. S., Casey, A., Orr-Ewing, A. J. & Reid, J. P. Selection and characterization of aerosol particle size using a Bessel beam optical trap for single particle analysis. *Physical Chemistry Chemical Physics* **14**, 6741-6748, doi:10.1039/C2CP40371D (2012).
- 37 David, G., Esat, K., Thanopoulos, I. & Signorell, R. Digital holography of optically-trapped aerosol particles. *Communications Chemistry* **1**, 46, doi:10.1038/s42004-018-0047-6 (2018).
- 38 Xu, W., Jericho, M. H., Meinertzhagen, I. A. & Kreuzer, H. J. Digital in-line holography of microspheres. *Applied Optics* **41**, 5367-5375, doi:10.1364/AO.41.005367 (2002).
- 39 Davies, N. W. *et al.* Evaluating biases in filter-based aerosol absorption measurements using photoacoustic spectroscopy. *Atmos. Meas. Tech.* **12**, 3417-3434, doi:10.5194/amt-12-3417-2019 (2019).
- 40 Dholakia, K., Spalding, G., & MacDonald, M. Optical tweezers: The next generation. *Physics World* **15(10)**, 31-35, doi:<https://doi.org/10.1088/2058-7058/15/10/37> (2002).
- 41 Rafferty, A., Gorkowski, K., Zuend, A. & Preston, T. C. Optical deformation of single aerosol particles. *Proceedings of the National Academy of Sciences* **116**, 19880-19886 (2019).
- 42 Prodi, F. *et al.* Digital holography for observing aerosol particles undergoing Brownian motion in microgravity conditions. *Atmospheric research* **82**, 379-384 (2006).
- 43 Berg, M. J. & Videen, G. Digital holographic imaging of aerosol particles in flight. *Journal of Quantitative Spectroscopy Radiative Transfer* **112**, 1776-1783 (2011).
- 44 Kempainen, O., Laning, J. C., Mersmann, R. D., Videen, G. & Berg, M. J. Imaging atmospheric aerosol particles from a UAV with digital holography. *Scientific Reports* **10**, 16085, doi:10.1038/s41598-020-72411-x (2020).
- 45 Bianco, V. *et al.* in *Fringe 2013*. (ed Wolfgang Osten) 911-916 (Springer Berlin Heidelberg).
- 46 Kim, J., Go, T. & Lee, S. J. Volumetric monitoring of airborne particulate matter concentration using smartphone-based digital holographic microscopy and deep learning. *Journal of Hazardous Materials* **418**, 126351, doi:<https://doi.org/10.1016/j.jhazmat.2021.126351> (2021).
- 47 Wu, Y.-C. *et al.* Air quality monitoring using mobile microscopy and machine learning. *Light: Science & Applications* **6**, e17046-e17046, doi:10.1038/lsa.2017.46 (2017).
- 48 Xu, W., Jericho, M., Meinertzhagen, I. & Kreuzer, H. Digital in-line holography for biological applications. *Proceedings of the National Academy of Sciences* **98**, 11301-11305 (2001).
- 49 Brunnhofer, G., Bergmann, A., Klug, A. & Kraft, M. Design and Validation of a Holographic Particle Counter. *Sensors (Basel)* **19**, 4899, doi:10.3390/s19224899 (2019).
- 50 Laning, J. C. & Berg, M. J. Orthographic imaging of free-flowing aerosol particles. *OSA Continuum* **2**, 3514-3520, doi:10.1364/OSAC.2.003514 (2019).

Reviewers' comments:

Reviewer #1 (Remarks to the Author):

All the comments were well reflected in the revised manuscript, including additional experiments and explanations.

Reviewer #4 (Remarks to the Author):

Review for "Advancing the Science of Dynamic Airborne Nanosized Particles using Nano-DIHM"

This revised article considers a variety of in line digital holography experiments aimed at establishing the capability to image sub-micron particles that are moving, that is, not trapped or deposited. Specifically, the authors claim that they characterize the size and shape of moving particles, their trajectory when they move; and all for particles as small as 100-200 nm both spherical and nonspherical in shape. I will note that I am not an original reviewer of this work, and the revised manuscript is the first time I am seeing the work.

The authors are correct that an ability to resolve the size and shape of sub-micron particles, especially as small as 100 nm, would be a major advance for aerosol science. This would be true for any imaging method that could work for moving particles. And so, I am excited about this work and offer the authors encouragement. However, after seeing the data and reading the other reviewers' comments, it is clear to me that their claim of image resolution at the 100-200 nm scale is questionable and not well defended.

The authors are correct that digital holography has been applied to moving aerosol particles in the past and that the particle size resolved in those cases is greater than one micron. Indeed, there is good reason for this larger resolution size. When particles are in motion, such as with an aerosol, one can usually only record a single hologram before the particles are out of the field of view. This means that the image resolution will be constrained by the normal restrictions of classical digital holography, and exclude the possibility of using methods like pixel super-resolution digital holography, which requires multiple hologram recordings. When this is understood, I know of no way that a resolution smaller than the diffraction limit can be achieved. So, it is certainly a pleasant surprise that I see, e.g., Fig. 1(f), which looks so good, I thought it was a microscope image! But then I read in relation to this figure, "The reconstruction of the cropped area (c1) in Figure 1 (c), as displayed in Figure 1(d), is of interest. Figure 1(e) is an example of high-resolution images of Figure 1(d); this high resolution is achieved by performing in-focus reconstruction. Figure 1(f) is a zoomed-in area of Figure 1(e)." So, is Fig. 1(d) in focus? Or is it not and Figs. 1(d)-1(f) are? Certainly, if Fig. 1(f) is really a holography image, it is among the best resolution I have ever seen. However, the authors are incorrect that these particles are moving aerosol particles. Rather they are particles that have deposited, i.e., stuck, to the side of the cuvette and are thus stationary. I know this because the authors report a *single* reconstruction (z) distance for *all* the particles 1 and 2 in the image. There is almost no chance that flowing particles would randomly be in a single plane (z distance) unless they have stuck to a surface. The same appears true in Fig. 3.

Then I come to Fig. 4 where particles fixed on a microscope slide are considered and I begin to really question the claims of sub-micron resolution. For example, Fig. 4(g) shows an S/TEM image of the

particles where a box outlines a group of 100 nm particles. A group of features is then shown in Fig. 4(h) and is associated with the particles in the box S/TEM image in Fig. 4(g). Obviously, they are **not** the same particles and if they are, the resolution is clearly too poor to resolve 100 nm. Nevertheless, it seems that in the holographic image, Fig. 4(h), there are single dots that could possibly be single particles. But then I notice the absence of background noise in this image. In my experience in digital holography, such little noise is only obtained by filtering and/or binarizing the image. If this is being done, and I suspect it is, then that filtering may be providing the illusion that single particles at 100 nm are being resolved. This is because even a blurry image of a 100 nm particle – what I would expect in this case – would appear after filtering to have crisp resolution. I see this kind of filtering being done throughout the work.

Finally, I come to Fig. 8 where the measurement configuration is shown. There is nothing notably new in this setup that could explain the resolution performance. The argument that allowing the hologram to expand, or zoom, before being recorded will not generate any improved resolution because the solid angle of the interference pattern only **decreases** as the sensor is taken farther from the particle in the zoom process. Having the particle very close to the pinhole is helpful in comparison to other approaches, but it introduces no new physical mechanism that could beat the diffraction limit. Of course, the pinhole itself will diffract! I looked up the references that the authors cite when they say, “studies using near-field optical microscopy have successfully captured the nanoscale objects, overcoming the diffraction limit of the optical system.” These references relate to either NSOM techniques or super-resolution fluorescence techniques. None of them have any relevance to holography as far as I can tell. So, I am not convinced and wonder why the authors think these papers support their work. For all of these reasons, I do not think that this work is ready to publish with the claims of sub-micron resolution as they are. Moreover, I do not think this is a matter of simply revising the work again and resubmitting essentially the same results. There needs to be real and clear evidence that 100-200 nm nonspherical particles are resolved **without** filtering the images and in cases where it is clear that the particles are flowing and **not** simply deposited on the cuvette. Then, the authors would have a dynamite paper.

Prof. Parisa Angeline Ariya

Department of Chemistry and Department of Atmospheric and Oceanic Sciences

McGill University, 801 Sherbrooke St. West Montreal, PQ, Canada

RE: Revision of the manuscript entitled "Advancing the Science of Dynamic Airborne Nanosized Particles using Nano-DIHM"

September 24, 2021

To: To: Editor/reviewers,
Communications Chemistry: Nature

Dear editor/colleagues,

Please find attached the revised manuscript entitled "Advancing the Science of Dynamic Airborne Nanosized Particles using Nano-DIHM" by Devendra Pal, Yevgen Nazarenko, Thomas C. Preston, and Parisa A. Ariya, for consideration for publication in your journal.

We would like to thank all reviewers for their time and consideration sincerely. We very much appreciate your critical comments and suggestions. We have considered them all and have made the requested significant modifications to the manuscript accordingly. We have also performed additional analysis and improving the quality of data on two fronts: 1) targeted experiments to provide the evidence of nanosized particles, and 2) improvement of image reconstructions and temporal resolutions, providing solid evidence for Nano-DIHM being an in-situ and real-time aerosols size, shape and phase observation of all sizes including *airborne nanoparticles trajectories*.

- **The combination of three concurrent processes** has been used in study to achieve a resolution smaller than the diffraction limit. 1) Improved experimental approach, 2) Numerical reconstruction process and 3) several holograms recorded in aqueous medium.
- The major conclusion of reviewer 4 was "*There needs to be real and clear evidence that 100-200 nm nonspherical particles are resolved *without* filtering the images and in cases where it is clear that the particles are flowing and *not* simply deposited on the cuvette*". We believe that there might be a slight misunderstanding as:

- (1) We do not use any image filtering. Instead, we used deconvolution routines, and
- (2) We have already demonstrated the airborne dynamic nanosize particle detection (Figures 1-4). Yet, we included further data for airborne 100 nm PSL *trajectories* in the revised manuscript that conclusively demonstrate that airborne nanoparticles are in motion (i.e., dynamic; Figure 4, movies Giff S1 and S2) and not "*simply deposited on the cuvette*", as kindly stated by the reviewer.

We have thus included a response in the response letter and the revised manuscript for your reconsideration. We believed we have addressed the point-to-point concerns of reviewer 4. Since reviewer 4 raised excellent points, we decided to revise the manuscript to provide much more succinct evidence that will facilitate the understanding of this innovation.

Thanks very much in advance for your time and reconsideration. If you require any further information, please feel free to contact me.

Cordially,

Parisa A. Ariya

James McGill Chair of Chemistry and Atmospheric and Oceanic Sciences

Reply

For your convenience, the reviewer's comments are italicized, and our responses are in regular font. We would like to cordially thank reviewers 1 and "4" and the editor for their time, contributions, and comments. We truly appreciate them, and we trust that they are satisfied with our modifications that address all their concerns.

Reviewers' comments:

Reviewer #1 (Remarks to the Author):

"All the comments were well reflected in the revised manuscript, including additional experiments and explanations."

Response: Thank you very comments. We very much appreciate that your previous inputs. We are delighted that you appreciated the novelty and potential in aerosol sciences.

Reviewer #4 (Remarks to the Author):

"Review for "Advancing the Science of Dynamic Airborne Nanosized Particles using Nano-DIHM"

This revised article considers a variety of in line digital holography experiments aimed at establishing the capability to image sub-micron particles that are moving, that is, not trapped or deposited. Specifically, the authors claim that they characterize the size and shape of moving particles, their trajectory when they move; and all for particles as small as 100-200 nm both spherical and nonspherical in shape. I will note that I am not an original reviewer of this work, and the revised manuscript is the first time I am seeing the work. The authors are correct that an ability to resolve the size and shape of sub-micron particles, especially as small as 100 nm, would be a major advance for aerosol science. This would be true for any imaging method that could work for moving particles. And so, I am excited about this work and offer the authors encouragement. However, after seeing the data and reading the other reviewers' comments, it is clear to me that their claim of image resolution at the 100-200 nm scale is questionable and not well defended".

Response: Thank you very much for your appreciation and positive remarks towards advancing this technique in aerosols science. We would also like to thank you for your critical comments and suggestions. We know that they are very constructive and appreciate it.

We herein addressed all your comments, explained slightly misunderstandings, and provided additional evidence for airborne particles in motion, namely trajectories of 100 nm airborne PSL particles (Figure 4), which conclusively demonstrate the dynamic tracking of airborne nanosized particles. For your convenience, we have enclosed additional movies (Giff S1 and S2), and Supplementary Information Figure S11 clearly shows the small changes in Z position changes the particles from focus to defocused.

"The authors are correct that digital holography has been applied to moving aerosol particles in the past and that the particle size resolved in those cases is greater than one micron. Indeed, there is good reason for this larger resolution size. When particles are in motion, such as with an aerosol, one can usually only record a single hologram before the particles are out of the field of view. This means that the image resolution will be constrained by the normal restrictions of classical digital holography and exclude the possibility of using methods like pixel super-resolution digital holography, which requires multiple hologram recordings. With this understood, I know of no way that a resolution smaller than the diffraction limit can be achieved."

Response: Thank you for your comment. Yes, indeed, you are absolutely, correct. In this study, we did not use the pixel super-resolution approach. Instead, we used **the combination of three concurrent processes** to achieve a resolution smaller than the diffraction limit.

Please kindly note that several other researchers have chosen not to proceed with the pixel super-resolution approach¹⁻²², which you have correctly referred to in your comment. Instead, we applied the numerical reconstruction process based on the Kirchhoff–Helmholtz transform algorithm with multiple deconvolution processes to achieve high resolution^{6,8,14}, described in detail in the method section in the revised manuscript.

In brief, **the combination of three concurrent processes** that have been used to achieve a better lateral resolution than the diffraction limits of the system are:

1) Improved experimental approach:

- In this work, we do not need to add any additional physical objects, and **there is no need for external filtering** for the recording/reconstruction of holograms^{9,11,23,24}.
- The holograms were recorded near the pinhole and bring the camera closer to the quartz flow tube cuvette for dynamic mode or quartz microscopic slide for stationary mode.
- This configuration is **possible because the Octopus software**²⁴ still allows us to **record the hologram** in the size of 2048 × 2048-pixel with a pixel size of 5.5 μm given the source to camera distance (5 mm). Hence, we achieved a larger field of view in this study, **allowing tracking both single-particle and multiple particles**.
- The **pinhole and the flow tube (cuvette) for dynamic experiments** or **microscopy slides for the stationary** experiment nearly touched each other. In DIHM, the shorter the source-to-sample distance, the higher the magnification, and hence a higher resolution can be achieved^{9,11,25}. By using this process, the numerical aperture (NA) of the Nano-DIHM increases substantially (NA = 0.7428) (Eqn. 3) in the revised manuscript).

$$NA = \frac{W}{2 \left[\sqrt{\left(\frac{d}{2}\right)^2 + (d)^2} \right]}, \quad \text{((Eqn. 3) in the revised manuscript)}$$

Where W is the width of the screen and d is the distance between the point source and the screen. Thus, **the theoretical lateral resolution of the Nano-DIHM is 271 nm, and after following the steps below, the nano-DIHM resolution can go below this theoretical threshold**^{9-12,14,25}.

- *Keeping the cuvette near the source **and** bringing the camera to near the cuvette **collectively increases the Nano-DIHM magnification***^{9,11}. Previously, the only way to increase the resolution was by numerical approaches because the low numerical aperture was one of the major issues in achieving the high resolution in digital holography^{9,12,14}.
- Another advantage of the current experimental setup shown in Figure 8(a) is that the single pinhole can work with **multiple virtual illumination sources**. Thus, we do not need multiple illumination sources to record the hologram.
- To obtain such short distances, the sample flow tube is introduced between the pinhole and CMOS so that the sample is facing the pinhole, as illustrated in Fig. 8 (a, b). The pinhole emits the light, and the bottom of the surface of the sample carrier will partially reflect this light to the pinhole. In turn, this light is reflected to the sample carrier, where it is partially reflected backward, again, and so on. As a result, the light coming directly from the pinhole will be superposed upon the reflected light, which appears to come from **several virtual pinholes**²⁵.
- This experimental recording process of holograms enabled us to overcome the diffraction barrier, allowing **measurement of the nanosized particles ≤ 200 nm, as shown in the revised Figures 1-4**.
- Note that several studies using near-field optical microscopy have successfully captured the nanoscale objects, overcoming the diffraction limit of the optical system²⁶⁻³². In digital holography, recent studies achieved detection of nanoparticles using an on-chip microscope where each nanoparticle served as a nucleus for self-assembled deposition of refractive materials (Polyethylene Glycol (PEG)-based solution) around each particle (nanolenses), thus increasing each particle's size and scattering cross-section, effectively helping it's the detection on a chip^{20,33-37}. Several theoretical numerical methods based on the **deconvolution algorithm on the sensor chip and point source** are used to detect **the nanoparticles**^{19,21,38-40}.
- In this study, in **Figure S14**, we have tested PEG, as previously reported in the literature, to confirm the refractive index measurements (SI, Figure S14 and Text S2).
- As the distance between the source and the sample gets shorter, the object vibrations and noise are getting more extensive due to the higher magnification^{1,4}, as expressed correctly by reviewer 4. It blurs out the fine interference fringes and reduces the potentially achievable high resolution. Indeed, it is a challenge, and we agree with it. However, experimentally, to overcome this significant challenge and compensate for the vibrations and noise, a short-time acquisition sequence of the holograms and a thin sample carrier (compared to the distance between the pinhole and the image sensor) was used and recorded^{3,4}. **Such numerical reconstruction methods with deblurring techniques** have been demonstrated that in the motion-deblurring photo-image analysis permits achieving a **higher resolution by a factor of 2 or more**^{1,4,24}.

2) Numerical reconstruction process:

- The Octopus/Stingray software^{23,24} was used to reconstruct the recorded holograms in this study. The software is based on a patented algorithm⁶, which can effortlessly achieve the lateral resolution on the order of half-wavelength ($\lambda/2$) of source and depth resolution on the order of one wavelength (λ)¹⁴. Furthermore, as shown in detail elsewhere,^{1,4,10,14} **a higher resolution** was achieved using **multiple deconvolutions routines** during the reconstruction: **(1) illumination system (pinhole) and (2) the finite numerical aperture of the recording screen (CCD/CMOS)**^{1,4,10,14}.

- Implementing an **instant 3D-deconvolution routine in our reconstruction** method allowed us to reach the desire resolution ^{2,4,10,14,24}. The importance is that to find the plane where the phase/intensity image is accurately focused. Thereby, if we are aimed to measure the PSL for 200 nm size, as an example, the plane must be within 0.01 micron. Otherwise, the dots will be only a few pixels and do not look like quality images. For that reason, increasing the precision with 0.001 in Octopus software allowed us to achieve high resolution.
- Finally, we accurately focus on blurred objects by adjusting reconstruction position (z) to up to three decimal places (0.001).
- Please note that as we have put in the acknowledgement section, *Dr. Hans Kreuzer*, the co-inventor of the digital inline holographic microscopy, guided us to improve the resolution in the Octopus software.
- The advantage of the deconvolution routines is to remove white noise from final reconstructed images and enhance the resolution by a factor of 2, as discussed in detail in previous studies ^{1,2,4,14}.
- By employing the **3D-deconvolution routines**, the out-of-focus signal is brought back to its scatterer, and the twin images are automatically removed as they are not part of the scattered wave. Thus, spatially well-localized parts of an object are recovered free from artifacts and noise-free. For example, Nickerson et al. ¹⁴ in Figure 5 show how implementing the two-fold deconvolution mechanism removes the white noise from reconstructed images.

3) Aqueous medium

- In addition to air, several holograms are formed in an aqueous medium (water), and thus the wavelength of the laser is reduced ^{9,12} from $\lambda = 405$ nm to ~ 300 nm.

Hence the combination of 3 concurrent steps is the procedure to go beyond the classical diffraction limit. In the revised manuscript, you will find the detailed information (section B) of the combination of three concurrent steps taken to provide high-resolution imaging in the Methods (Lines from 536-606).

*"So, it is certainly a pleasant surprise that I see, e.g., Fig. 1(f), which looks so good, I thought it was a microscope image! But then I read in relation to this figure, "The reconstruction of the cropped area (c1) in Figure 1(c), as displayed in Figure 1(d), is of interest. Figure 1(e) is an example of high-resolution images of Figure 1(d); this high resolution is achieved by performing in-focus reconstruction. Figure 1(f) is a zoomed-in area of Figure 1(e)." So, is Fig. 1(d) in focus? Or is it not and Figs. 1(d)-1(f) are? Certainly, if Fig. 1(f) is really a holography image, it is among the best resolution I have ever seen. However, the authors are incorrect that these particles are moving aerosol particles. Rather they are particles that have deposited, i.e., stuck, to the side of the cuvette and are thus stationary. I know this because the authors report a *single* reconstruction (z) distance for *all* the particles 1 and 2 in the image. There is almost no chance that flowing particles would randomly be in a single plane (z distance) unless they have stuck to a surface. The same appears true in Fig. 3."*

Response: Thanks for your comments. Please note that we have already provided multiple evidence for airborne 100 nm PSL particles in the dynamic modes. However, for further clarity, we have chosen our words more accurately to avoid any ambiguity.

We have already shown multiple evidence for airborne 100 nm PSL particles. The following steps have been taken to achieve the high-quality images in the manuscript. These are:

- **A raw hologram** (Figure 1 a) contains information for 100 nm airborne PSL particles, i.e., in a dynamic airflow with a flow rate of 1.7 L/min.
- **A background hologram** (Figure 1 b) was recorded for the dry air purified using three HEPA filters, leading to a particle-free spectrum.
- **A contrast hologram** (Figure 1 c) resulting from the subtraction of the background hologram from the raw hologram. Subtracting the background from the raw hologram removes the possible contamination due to the source (pinhole imperfections) and the object holder (a cuvette or microscopy slide). Thus, **this process significantly improves the reconstruction of an image, as discussed in numerous publications**^{7,9,11}.
- Figure 1d is the zooming area of the circle shown in Figure 1c.
- In figure 1 e, we performed deconvolution and in-focus reconstruction, enhancing image quality and reducing noise^{1,4,5,7,9,10,14,41}.
- Figure 1f is a zoom-in area for Figure 1 e, and adjusted the focus for reconstruction; as such, we obtained high-resolution quality of Figure 1f.
- The reconstructed results shown in **Figure 1f are indeed for moving aerosols**. During the hologram recording process, we pass the aerosol stream, which consists of multiple particles (see Fig. 1 (m & n)), and thus even **a single hologram can carry the information for many moving particles**.
- **During the numerical reconstruction process, many particles can be in focus at a particular Z value**.
- Previous literature has shown similar results at particular Z values, and many particles were detected^{7,9,11,12,33,34,42}. For example, Xu et al.⁴³ in Figures 1 and 2 both showed that several particles trajectories were detected at a given Z value. Also, Garcia et al.⁹ in Figure 5 showed that several particles could be detected at a particular Z value.
- The revised manuscript provided the particle's trajectory within 62.5-millisecond intervals for several hologram acquisitions in SI-movie Giff 1 and SI-movie Giff 2. We also provided **Figure 4, showing the motion of PSL particles at the same reconstruction position in multiple holograms (total 16 and 9) within the temporal resolution at 62.5 ms**.

Figure 4 Trajectories of 100 nm airborne PSL particles in motion within the flow tube cuvette: (a) Sum of 16 holograms with 62.5 ms temporal resolution. (b) The intensity response of selected particles and their diameters are 180, 110 nm (FWHM), respectively. (c) Sum of 9 holograms with 62.5 ms temporal resolution. (d) reconstruction of summed hologram (c) in one plane with the trajectory of circled data in focus. Red arrows indicate the directions of particles motion.

- The dynamic 4D trajectories (3D positions and 1D time) of 100 nm PSL spheres are provided in Figure 4 (a-d), movie-SI-Giff 1 and movie-SI-Giff 2 movies with a temporal resolution of 62.5 ms. The SI-Giff 1 is a zoomed-in image of SI-Giff 2.
- To obtain high-resolution trajectories of the airborne nanoparticles in this study, we used the following procedure:
 - (1) A sequence of holograms was recorded at 16fps, thus 62.5 ms-time intervals.
 - (2) The background contaminations were eliminated by subtraction of consecutive holograms, and
 - (3) The resultant holograms were reconstructed at a given plane and summed to obtain the final trajectories (16 holograms summed in Figure 4 (a) and 9 holograms summed in Figure 4(c)). Subtracting holograms ensured that the dynamic range was not exceeded and only the object-related information (moving PSL particles information) was retained.

- Figure 4 (a & c) shows the sequential positions at successive recording times of the airborne 100 nm PSL particles contained in the sample volume in two reconstruction planes separated by 200 μm . Figure 4 (b) shows the zoomed-in crop area of trajectories given in Figure (Figure 4(a)) to provide the confirmation of the size of 100 nm PSL airborne particles. Sixteen sequential positions clearly define the trajectory of airborne PSL particles (Figure. 4(a)), which moving in a random fashion more or less confined to a plane parallel to the flow tube cuvette.
- All the reconstructions of the holograms were performed at the same reconstruction distance (z) and then processed to create the Giff movies. As depicted in SI-Giff 1, the dark red particles are in the focus of the reconstruction plane, while some of the particles (green/blue) are a little bit out of focus at the exact reconstruction distances due to the finite depth of field of the objective. As seen in Figure 4 (a-d), some PSL particles are in the focus reconstruction plane, but some are progressively out of focus, indicating that the motion direction also has a component perpendicular to the reconstruction plane. In order to overcome the out-of-focus reconstruction in-complete trajectories analysis, several reconstructions from the same hologram in as many planes are necessary^{9,43}. As such, we demonstrated the ability of the Nano-DIHM to visualize particle trajectories."

In the revised manuscript, the above description is included in the results section from Line 256-294.

*"Then I come to Fig. 4 where particles fixed on a microscope slide are considered and I begin to really question the claims of sub-micron resolution. For example, Fig. 4(g) shows an S/TEM image of the particles where a box outlines a group of 100 nm particles. A group of features is then shown in Fig. 4(h) and is associated with the particles in the box S/TEM image in Fig. 4(g). Obviously, they are *not* the same particles and if they are, the resolution is clearly too poor to resolve 100 nm. Nevertheless, it seems that in the holographic image, Fig. 4(h), there are single dots that could possibly be single particles.*

Response: Thank you for your comment. Yes, the dots present in previous Figure 4(h) (now, Figure 5h) are indeed single PSL particles.

But then I notice the absence of background noise in this image. In my experience in digital holography, such little noise is only obtained by filtering and/or binarizing the image. If this is being done, and I suspect it is, then that filtering may be providing the illusion that single particles at 100 nm are being resolved. This is because even a blurry image of a 100 nm particle – what I would expect in this case – would appear after filtering to have crisp resolution. I see this kind of filtering being done throughout the work."

Response: Thank you for the excellent comment. We fully understand and appreciate your comment. Nevertheless, we did not use any external filtration of images in the manuscript. Indeed, there is insignificant noise, but it is not due to external filtering (see below).

Figure S11 Intensity reconstruction at different depths of PSL particles on a microscopy slide. It shows the small changes in Z position changes the particles from focus to defocused.

It is part of the deconvolution process, which is done in two steps.

- **First**, we subtracted the background from the raw hologram, which removes possible optical contaminations due to the source (pinhole imperfections) and the object holder (a flow tube cuvette or microscopy slide). As discussed in numerous publications in the scientific literature ^{7,9,11}, this process significantly improves image reconstruction.
- **Secondly**, a previous patented algorithm ⁶ that includes multiple deconvolution processes and a Wiener filter overcame the white noise from the final reconstructed images and enhanced the resolution by a factor 2 ^{1,2,4,14}. This is because of employing 3D-deconvolution methods, the out-of-focus signal is brought back to its scatterer, and the twin images are automatically removed as they are not part of the scattered wave. Thus, spatially well-localized parts of an object are recovered free from artifacts and noise-free. For example, Nickerson et al. [14] in Figure 5 show

how implementing the two-fold deconvolution mechanism removes the white noise from reconstructed images and produces the best image quality.

- The revised enclosed manuscript also provides an example of how small changes of Z values can significantly influence reconstruction quality (image quality/resolution) (Supplementary information, Figure S11). We also attached Figure S11 below for your convenience.
- Please kindly note that we are not the first ones who attempted this approach. Garcia et al.,^{9,11}, have shown an example of changing the reconstruction distance can alter the resolution of images; focusing on the crop images of particular interest of study has been shown significant improvement in image quality, as we have shown in this study (Figure S11) by changing reconstruction Z we have obtained image quality improvement.
- A recent study, Onur et al.,³⁴ showed that cropped images of research of interest have been helpful to optimize the image quality. This study particularly focused on detecting nanoparticles in static mode. Garcia et al.,^{9,11} showed an example of how changing the reconstruction distance could alter the resolution of images. Focusing on the cropped images of particular interest has significantly improved image quality, as we also showed in this study by changing Z (Figure S11).

*"Finally, I come to Fig. 8 where the measurement configuration is shown. There is nothing notably new in this setup that could explain the resolution performance. The argument that allowing the hologram to expand, or zoom, before being recorded will not generate any improved resolution because the solid angle of the interference pattern only *decreases* as the sensor is taken farther from the particle in the zoom process. Having the particle very close to the pinhole is helpful in comparison to other approaches, but it introduces no new physical mechanism that could beat the diffraction limit. Of course, the pinhole itself will diffract! I looked up the references that the authors cite when they say, "studies using near-field optical microscopy have successfully captured the nanoscale objects, overcoming the diffraction limit of the optical system." These references relate to either NSOM techniques or super-resolution fluorescence techniques."*

Response: Thank you for your thoughts and comments.

Firstly, Figure 8 (Now Figure 9) is the schematical of the experimental setup. In the revised manuscript, it has been revised slightly to address your comments. Please kindly note that it is for intellectual transparency reasons for all leading publishers, including Nature publishing journals.

As your major comment on the performance we have addressed above, yet we herein reiterate for your convenience. Sorry in advance for the reiteration, we believe that it facilitates the communication.

Improved experimental approach:

- In this study, we do not need to add any additional physical objects, or **there is no need for external filtering** for the recording/reconstruction of holograms^{9,11,23,24}.
- The holograms were recorded near the pinhole and bring the camera closer to the quartz flow tube cuvette for dynamic mode or quartz microscopic slide for stationary mode.
- This configuration is **possible because the Octopus software**²⁴ still allows us to **record the hologram** in the size of 2048 × 2048-pixel with a pixel size of 5.5 μm given the source to

camera distance (5 mm). Hence, we achieved a larger field of view in this study, **allowing tracking both single-particle and multiple particles**.

- The **pinhole and the flow tube (cuvette) for dynamic experiments** or **microscopy slides for the stationary** experiment nearly touched each other. In DIHM, the shorter the source-to-sample distance, the higher the magnification, and hence a higher resolution can be achieved^{9,11,25}. By using this process, the numerical aperture (NA) of the Nano-DIHM increases substantially (NA = 0.7428) (Eqn. 3) in the revised manuscript). The theoretical lateral resolution of the Nano-DIHM is 271 nm, and after following the steps below, the nano-DIHM resolution can go below this theoretical threshold^{9-12,14,25}.
- *Keeping the cuvette near the source **and** bringing the camera to near the cuvette collectively increases the Nano-DIHM magnification*^{9,11}. Previously, the only way to increase the resolution was by numerical approaches because the low numerical aperture was one of the major issues in achieving the high resolution in digital holography^{9,12,14}.
- Another advantage of the current experimental setup shown in Figure 8(a) is that the single pinhole can work with **multiple virtual illumination sources**. Thus, we do not need multiple illumination sources to record the hologram.
- To obtain such short distances, the sample flow tube is introduced between the pinhole and CMOS so that the sample is facing the pinhole, as illustrated in Fig. 8 (a, b). The pinhole emits the light, and the bottom of the surface of the sample carrier will partially reflect this light to the pinhole. In turn, this light is reflected to the sample carrier, where it is partially reflected backward, again, and so on. As a result, the light coming directly from the pinhole will be superposed upon the reflected light, which appears to come from **several virtual pinholes**²⁵.
- This experimental recording process of holograms enabled us to overcome the diffraction barrier, allowing **measurement of the nanosized particles ≤ 200 nm, as shown in the revised Figures 1-4**.
- Note that several studies using near-field optical microscopy have successfully captured the nanoscale objects, overcoming the diffraction limit of the optical system²⁶⁻³². In digital holography, recent studies achieved detection of nanoparticles using an on-chip microscope where each nanoparticle served as a nucleus for self-assembled deposition of refractive materials (Polyethylene Glycol (PEG)-based solution) around each particle (nanolenses), thus increasing each particle's size and scattering cross-section, effectively helping it's the detection on a chip^{20,33-37}. Several theoretical numerical methods based on the **deconvolution algorithm on the sensor chip and point source** are used to detect **the nanoparticles**^{19,21,38-40}.
- In this study, in **Figure S14**, we have tested PEG, as previously reported in the literature, to confirm the refractive index measurements (SI, Figure S14 and Text S2).
- As the distance between the source and the sample gets shorter, the object vibrations and noise are getting more extensive due to the higher magnification^{1,4}, as expressed correctly by reviewer 4. It blurs out the fine interference fringes and reduces the potentially achievable high resolution. Indeed, it is a challenge, and we agree with it. However, experimentally, to overcome this significant challenge and compensate for the vibrations and noise, a short-time acquisition sequence of the holograms and a thin sample carrier (compared to the distance between the pinhole and the image sensor) was used and recorded^{3,4}. Such numerical reconstruction methods with deblurring techniques have been demonstrated that in the motion-deblurring photo-image analysis permits achieving a higher resolution by a factor of 2 or more^{1,4,24}.

As for your last comment, we agree with your remark on the references that we have shown, and thus we have modified and added new ones related to holography in the reference section, thus removing any ambiguity and enhancing the clarity in the Methods and introduction section.

In the revised manuscript, the revised experimental procedure is now included in the method section from Line 538-606.

*"Moreover, I do not think this is a matter of simply revising the work again and resubmitting essentially the same results. There needs to be real and clear evidence that 100-200 nm nonspherical particles are resolved *without* filtering the images and in cases where it is clear that the particles are flowing and *not* simply deposited on the cuvette. Then, the authors would have a dynamite paper."*

Response: Thank you for your remark. We fully understand and appreciate your comments. In this response, we wish to address exactly what you have requested:

*" There needs to be real and clear evidence that 100-200 nm nonspherical particles are resolved *without* filtering the images and in cases where it is clear that the particles are flowing and *not* simply deposited on the cuvette":*

1. We do not use any image filtering. Instead, we used deconvolution routines. We have herein presented in detail three concurrent steps that allowed us to reach high resolution without any image filtering. We also incorporated them in the revised manuscript (Methods, lines 537 to 602).
2. Please kindly note that we have already demonstrated the airborne dynamic nanosize particle detection (Figures 1-4). Yet, we included further airborne 100 nm PSL *trajectories* in the revised manuscript that conclusively demonstrate that airborne nanoparticles are in motion (i.e., dynamic; Figure 4, movies Giff S1 and S2) and not "*simply deposited on the cuvette*", as you have requested (see below).

Figure 4 Trajectories of 100 nm airborne PSL particles in motion within the flow tube cuvette: (a) Sum of 16 holograms with 62.5 ms temporal resolution. (b) Intensity response of selected

particles and their diameters are 180, 110 nm (FWHM) respectively. (c) Sum of 9 holograms with 62.5 ms temporal resolution. And (d) reconstruction of summed hologram (c) in one plane with the trajectory of circled data in focus. Red arrows indicate the directions of particles motion.

We thus address your second comment that demonstrate the ability of the Nano-DIHM to track airborne nano-size particle trajectories (revised manuscript, Lines 256-294).

We would like to cordially thank both reviewers and the editor for their time, contributions, and comments. We truly appreciate them, and we trust that they are satisfied with our modifications that address all their concerns.

References used in this reply

- 1 Latychevskaia, T. & Fink, H.-W. Resolution enhancement in in-line holography by numerical compensation of vibrations. *Optics express* **25**, 20109-20124 (2017).
- 2 Latychevskaia, T. & Fink, H.-W. Holographic time-resolved particle tracking by means of three-dimensional volumetric deconvolution. *Optics Express* **22**, 20994-21003, doi:10.1364/OE.22.020994 (2014).
- 3 Germann, M., Latychevskaia, T., Escher, C. & Fink, H.-W. Pulsed electron holography. *Applied Physics Letters* **102**, 203115 (2013).
- 4 Latychevskaia, T., Gehri, F. & Fink, H.-W. Depth-resolved holographic reconstructions by three-dimensional deconvolution. *Optics Express* **18**, 22527-22544, doi:10.1364/OE.18.022527 (2010).
- 5 Xu, W., Jericho, M., Meinertzhagen, I. & Kreuzer, H. Digital in-line holography for biological applications. *Proceedings of the National Academy of Sciences* **98**, 11301-11305 (2001).
- 6 Kreuzer, H. J. US. Patent 6411406 B1, Canadian Patent CA 2376395. (2002).
- 7 Xu, W., Jericho, M. H., Meinertzhagen, I. A. & Kreuzer, H. J. Digital in-line holography of microspheres. *Applied Optics* **41**, 5367-5375, doi:10.1364/AO.41.005367 (2002).
- 8 Jericho, S., Garcia-Sucerquia, J., Xu, W., Jericho, M. & Kreuzer, H. Submersible digital in-line holographic microscope. *Review of scientific instruments* **77**, 043706 (2006).
- 9 Garcia-Sucerquia, J. et al. Digital in-line holographic microscopy. *Applied optics* **45**, 836-850 (2006).
- 10 Kanka, M., Riesenberger, R. & Kreuzer, H. J. Reconstruction of high-resolution holographic microscopic images. *Optics Letters* **34**, 1162-1164, doi:10.1364/OL.34.001162 (2009).
- 11 Jericho, M. H. & Kreuzer, H. J. in *Coherent Light Microscopy* 3-30 (Springer, 2011).
- 12 Jericho, M. H., Kreuzer, H. J., Kanka, M. & Riesenberger, R. Quantitative phase and refractive index measurements with point-source digital in-line holographic microscopy. *Applied Optics* **51**, 1503-1515, doi:10.1364/AO.51.001503 (2012).
- 13 Maciel, D., Veres, S. P., Kreuzer, H. J. & Kreplak, L. Quantitative phase measurements of tendon collagen fibres. *J Biophotonics* **10**, 111-117, doi:10.1002/jbio.201500263 (2017).
- 14 Nickerson, B. S. & Kreuzer, H. J. Deconvolution for digital in-line holographic microscopy. *Advanced Optical Technologies* **2**, 337-344, doi:doi:10.1515/aot-2013-0030 (2013).
- 15 Sánchez-Ortiga, E., Doblas, A., Saavedra, G., Martínez-Corral, M. & Garcia-Sucerquia, J. Off-axis digital holographic microscopy: practical design parameters for operating at diffraction limit. *Applied Optics* **53**, 2058-2066, doi:10.1364/AO.53.002058 (2014).
- 16 Singh, M. & Khare, K. Accurate efficient carrier estimation for single-shot digital holographic imaging. *Optics Letters* **41**, 4871-4874, doi:10.1364/OL.41.004871 (2016).
- 17 Eder, K. M., Marzi, A., Barroso, A., Kemper, B. & Schneckeburger, J. in *Imaging and Applied Optics Congress. JW5A.4* (Optical Society of America).

- 18 Verpillat, F., Joud, F., Desbiolles, P. & Gross, M. Dark-field digital holographic microscopy for 3D-tracking
of gold nanoparticles. *Optics Express* **19**, 26044-26055, doi:10.1364/OE.19.026044 (2011).
- 19 Dixon, L., Cheong, F. C. & Grier, D. G. Holographic deconvolution microscopy for high-resolution particle
tracking. *Optics Express* **19**, 16410-16417, doi:10.1364/OE.19.016410 (2011).
- 20 Ray, A. *et al.* Holographic detection of nanoparticles using acoustically actuated nanolenses. *Nature*
Communications **11**, 171, doi:10.1038/s41467-019-13802-1 (2020).
- 21 Warnasooriya, N. *et al.* Imaging gold nanoparticles in living cell environments using heterodyne digital
holographic microscopy. *Optics express* **18**, 3264-3273 (2010).
- 22 Fournier, C., Denis, L. & Fournel, T. Resolution in in-line Digital Holography. *Journal of Physics:*
Conference Series **206**, 012025, doi:10.1088/1742-6596/206/1/012025 (2010).
- 23 imaging, D. i. Stingray Software User Guide. (2018).
- 24 imaging, D. i. Octopus Software User Guide. (2018).
- 25 Kanka, M., Riesenber, R., Petruck, P. & Graulig, C. High resolution (NA=0.8) in lensless in-line
holographic microscopy with glass sample carriers. *Optics Letters* **36**, 3651-3653,
doi:10.1364/OL.36.003651 (2011).
- 26 Won, R. Eyes on super-resolution. *Nature Photonics* **3**, 368-369, doi:10.1038/nphoton.2009.103 (2009).
- 27 Zhuang, X. Nano-imaging with STORM. *Nature Photonics* **3**, 365-367, doi:10.1038/nphoton.2009.101
(2009).
- 28 Huang, B., Babcock, H. & Zhuang, X. Breaking the diffraction barrier: super-resolution imaging of cells.
Cell **143**, 1047-1058, doi:10.1016/j.cell.2010.12.002 (2010).
- 29 Liu, W. *et al.* Breaking the Axial Diffraction Limit: A Guide to Axial Super-Resolution Fluorescence
Microscopy. *Laser & Photonics Reviews* **12**, 1700333, doi:<https://doi.org/10.1002/lpor.201700333>
(2018).
- 30 Ozcan, A. *et al.* Differential near-field scanning optical microscopy. *Nano letters* **6**, 2609-2616 (2006).
- 31 de Lange, F. *et al.* Cell biology beyond the diffraction limit: near-field scanning optical microscopy. *J Cell*
Sci **114**, 4153-4160 (2001).
- 32 Betzig, E. & Chichester, R. J. Single Molecules Observed by Near-Field Scanning Optical Microscopy.
Science **262**, 1422, doi:10.1126/science.262.5138.1422 (1993).
- 33 Mudanyali, O., Bishara, W. & Ozcan, A. Lensfree super-resolution holographic microscopy using wetting
films on a chip. *Optics express* **19**, 17378-17389 (2011).
- 34 Mudanyali, O. *et al.* Wide-field optical detection of nanoparticles using on-chip microscopy and self-
assembled nanolenses. *Nature Photonics* **7**, 247-254, doi:10.1038/nphoton.2012.337 (2013).
- 35 Bishara, W. *et al.* Holographic pixel super-resolution in portable lensless on-chip microscopy using a
fiber-optic array. *Lab on a Chip* **11**, 1276-1279 (2011).
- 36 McLeod, E. *et al.* High-Throughput and Label-Free Single Nanoparticle Sizing Based on Time-Resolved
On-Chip Microscopy. *ACS Nano* **9**, 3265-3273, doi:10.1021/acsnano.5b00388 (2015).
- 37 McLeod, E. *et al.* Tunable Vapor-Condensed Nanolenses. *ACS Nano* **8**, 7340-7349,
doi:10.1021/nn502453h (2014).
- 38 Greenbaum, A. *et al.* Increased space-bandwidth product in pixel super-resolved lensfree on-chip
microscopy. *Scientific Reports* **3**, 1717, doi:10.1038/srep01717 (2013).
- 39 Wong, A., Kazemzadeh, F., Jin, C. & Wang, X. Y. Bayesian-based aberration correction and numerical
diffraction for improved lensfree on-chip microscopy of biological specimens. *Optics Letters* **40**, 2233-
2236, doi:10.1364/OL.40.002233 (2015).
- 40 Tippie, A. E., Kumar, A. & Fienup, J. R. High-resolution synthetic-aperture digital holography with digital
phase and pupil correction. *Optics Express* **19**, 12027-12038, doi:10.1364/OE.19.012027 (2011).
- 41 Micó, V., Zalevsky, Z., Ferreira, C. & García, J. Superresolution digital holographic microscopy for three-
dimensional samples. *Optics Express* **16**, 19260-19270, doi:10.1364/OE.16.019260 (2008).
- 42 Kempainen, O., Laning, J. C., Mersmann, R. D., Videen, G. & Berg, M. J. Imaging atmospheric aerosol
particles from a UAV with digital holography. *Scientific Reports* **10**, 16085, doi:10.1038/s41598-020-
72411-x (2020).

43 Xu, W., Jericho, M. H., Kreuzer, H. J. & Meinertzhagen, I. A. Tracking particles in four dimensions with in-line holographic microscopy. *Optics Letters* **28**, 164-166, doi:10.1364/OL.28.000164 (2003).

REVIEWERS' COMMENTS:

Reviewer #4 (Remarks to the Author):

[Editorial Note: This reviewer has not provided any further comments to the Authors.]

Prof. Parisa Angeline Ariya

Department of Chemistry and Department of Atmospheric and Oceanic Sciences

McGill University, 801 Sherbrooke St. West Montreal, PQ, Canada

RE: Revision of the manuscript entitled "Advancing the Science of Dynamic Airborne Nanosized Particles using Nano-DIHM"

November 9, 2021

To: To: Editor/reviewers,
Communications Chemistry: Nature

Dear editor/colleagues,

Thank you for accepting the manuscript entitled "Advancing the Science of Dynamic Airborne Nanosized Particles using Nano-DIHM" by Devendra Pal, Yevgen Nazarenko, Thomas C. Preston, and Parisa A. Ariya.

We would like to thank all reviewers for their time and consideration sincerely. We very much appreciate your critical comments and suggestions. We have considered them all and have made the requested significant modifications to the manuscript.

Thanks very much in advance for your time and reconsideration.

Cordially,

Parisa A. Ariya

James McGill Chair of Chemistry and Atmospheric and Oceanic Sciences

Reply

Reviewers' comments:

Reviewer # (Remarks to the Author):

"None"

Response: Thank you very much. We very much appreciate that your previous inputs. We are delighted that you appreciated the novelty and potential in aerosol sciences.